# Persistent carbon sink at a boreal drained bog forest

Kari Minkkinen[1], Paavo Ojanen[1], Timo Penttilä[2], Mika Aurela[3], Tuomas Laurila[3], Juha-Pekka Tuovinen[3] and Annalea Lohila[3]

[1] Department of Forest Sciences, P.O. Box 27, FI-00014 University of Helsinki, Finland
[2] Natural Resources Institute Finland, P.O. Box 2, FI-00791 Helsinki, Finland,
[3] Finnish Meteorological Institute, P.O. Box 503, 00101, Helsinki, Finland

*Correspondence to*: Kari Minkkinen (kari.minkkinen@helsinki.fi)

**Abstract**. Drainage of peatlands is expected to turn these ecosystems into carbon sources to the atmosphere. We measured carbon dynamics of a drained forested peatland in southern Finland over four years, including one with severe drought during growing season. Net ecosystem exchange (NEE) of carbon dioxide ($CO_2$) was measured with the eddy covariance method from a mast above the forest. Soil and forest floor $CO_2$ and methane ($CH_4$) fluxes were measured from the strips and from ditches with closed chambers. Biomass and litter production were sampled, and soil subsidence was measured by repeated levellings of the soil surface. The drained peatland ecosystem was a strong sink of carbon dioxide in all studied years. Soil $CO_2$ balance was estimated by subtracting the carbon sink of the growing tree stand from NEE, and it showed that the soil itself was a carbon sink as well. A drought period in one summer significantly decreased the sink through decreased gross primary production. Drought also decreased ecosystem respiration. The site was a small sink for $CH_4$, even when emissions from ditches were taken into account. Despite the continuous carbon sink, peat surface subsided slightly during the 10-year measurement period, which was probably mainly due to compaction of peat. It is concluded that even fifty years after drainage this peatland site acted as a soil C sink due to relatively small changes in water table and in plant community structure compared to similar undrained sites, and the significantly increased tree stand growth and litter production. Although the site is currently a soil C sink, simulation studies with process models are needed to test whether such sites could remain C sinks when managed for forestry over several tree-stand rotations.

## 1. Introduction

Peatlands worldwide contain 500–600 Pg carbon (C) (Gorham 1991, Yu et al. 2010, Page et al. 2011) that has been fixed from the atmosphere. Wet, anoxic conditions constrain the decomposition of organic matter and thus enable the accumulation of carbon as peat. Since wet conditions are a prerequisite for peat accumulation, drying of peatlands through drainage or climate change has been assumed to result in the release of sequestered carbon back to the atmosphere.

The effect of drainage of forested peatlands on carbon stocks has been under debate at least since the 1980's, when large carbon dioxide ($CO_2$) emissions were reported from drained peatlands in Finland (Silvola 1986). Studies from agricultural peat soils show that carbon stocks are usually greatly diminished under efficient drainage (e.g. Oleszczuk et al. 2008, Tiemeyer et al. 2016), with some exceptions (e.g. Merbold et al. 2009, Fleischer et al. 2016). Similar C loss has often been assumed for all drained peatlands, including those drained for forestry. However, in some peatlands, soil has been reported to sequester carbon even after drainage (e.g. Lohila et al. 2011, Turetsky et al. 2011). Minkkinen and Laine (1998a) and Minkkinen et al. (1999) showed, based on peat C stock measurements, that many nutrient-poor peatland sites remained C sinks after drainage. Later, Ojanen et al. (2013) showed the same relation with site type and soil C balance, nutrient poor ones being sinks and fertile ones sources. The continued C sequestration on relatively nutrient-poor sites have been related to the increased litter production and changes in litter quality (Laiho et al. 2003, Straková et al. 2012) vs. only moderately increased decomposition of old peat (Minkkinen and Laine 1998a). This view has, however, still been challenged (e.g. Simola et al. 2012), and, for example according to IPCC guidance, drained peatlands are assumed to be C sources (Drösler et al. 2014).

Climate warming, in addition to drainage has been predicted to increase C loss from peatlands because of increased soil temperatures and droughts (e.g. Moore 2002). In warmer and drier conditions the decomposition of soil organic matter (SOM) is expected to increase, although increased primary production and possible long-term shift towards more shrub and tree dominated vegetation communities (Laiho et al. 2003, Tahvanainen 2011, Straková et al. 2010, 2012) may partly compensate for the increased decomposition rates (Flanagan and Syed 2011). The reported impacts of droughts on ecosystem $CO_2$ fluxes are, however, variable. Droughts have been shown to decrease photosynthesis and increase ecosystem respiration especially on wet and nutrient-rich fens (Bubier et al. 2003, Adkinson et al. 2011), while on naturally drier bogs, the effects may be reversed (Sulman et al. 2010). $CO_2$ emissions from the decomposition of peat are often shown to increase linearly with water level drawdown (e.g. Silvola et al. 1996, Jauhiainen 2012) but there are also indications of an optimum water table depth in boreal peatlands below which soil respiration would not further increase (Mäkiranta et al. 2009). Thus, in some cases decomposition of SOM might even decrease during droughts (Sulman et al. 2010).

Our study site, the forestry-drained peatland Kalevansuo in South Finland, was earlier reported to be a strong C sink in terms of net ecosystem $CO_2$ exchange (NEE) during 2004–2005 (Lohila et al. 2011). The magnitude of the C sink was remarkably higher than the estimated tree stand C pool increment, which led us to the conclusion that also the soil must act as a C sink. Whether this was just a single-year result or whether it holds through several years with varying weather conditions, will be investigated in this paper.

The aims of this study were to estimate the full C balance of a drained peatland forest ecosystem over four years, and to analyse the impact of seasonal drought on the C fluxes. We measured the C pools in the ecosystem (peat soil, vegetation above and below ground), $CO_2$ fluxes between the ecosystem and atmosphere, namely NEE, gross primary production (GPP), ecosystem respiration ($R_{ECO}$) and forest floor respiration ($R_{FF}$) divided to component fluxes (peat, litter, roots, ground vegetation) and the C flux in litter (L). We complemented the results with measurements of methane ($CH_4$) fluxes and peat subsidence.

## 2. Material and Methods

### 2.1 Site

The measurements were carried out in a drained peatland forest, Kalevansuo, in southern Finland (60°38'49'' N, 24°21'23'' E, 123 m. a.s.l.). The peatland was drained by digging open ditches in 1971. Kalevansuo is a typical *dwarf shrub type* peatland forest according to the classification of Vasander and Laine (2008). The dominant tree species is Scots pine (*Pinus sylvestris* L.), comprising 98% of the stand volume and 53% of the stem number. Pubescent birch (*Betula pubescens* Roth) and Norway spruce (*Picea abies* L.) form the sparse understorey.

The site has been naturally forested long before drainage as evidenced by very old scattered stumps of Scots pine found in all parts of the site. Tree ages of the present Scots pine stand, as determined in 2005 from increment cores of sample trees (n=7), varied from 67 to 179 years with an average of 120 years. In 2008, the stand stem volume was 130 m³ ha⁻¹, basal area 18 m² ha⁻¹, dominant height 16 m and stem number 1670 ha⁻¹. Microtopographically the site is rather even (lawn level), with small hummocks covering about 25% of the area. A more detailed description of the stand is given by Lohila et al. (2011).

Following drainage, the tree stand has grown bigger and the coverage of mire species has decreased and forest species increased in the bottom and field layers. However, many mire species are still present at the peatland. Forest floor vegetation consists mainly of forest and mire dwarf shrubs (*Vaccinium myrtillus* L., *V. vitis idaea* L., *V. uliginosum* L., *Ledum palustre* L.), with patches of cottongrass (*Eriophorum vaginatum* L.) and cloudberry (*Rubus chamaemorus* L.). The dominant moss species are *Pleurozium schreberi* (Brid.) Mitt., covering 48 % of the study area, and *Dicranum polysetum* (37%), but *Sphagnum* mosses

such as *S. angustifolium* (Russ.) C. Jens., *S. russowii* Warnst., and *S. magellanicum* Brid. are also abundant in moist patches (coverage 15 %; Badorek et al. 2011). The ditches have not been cleaned since digging in 1971 and are nowadays totally vegetated, mainly with *Sphagnum riparium* (and *S. russowii, S. angustifolium*), some cottongrass (*Eriophorum vaginatum*) and sporadic dwarf shrubs (*Ledum palustre*).

Peat depth, measured from the 33 sample plots varies from 1.3 to 3.0 metres, average being 2.2 m. Mean peat bulk density is 94 kg $m^{-3}$ in the 0–20 cm layer. The peat accumulated prior to drainage at the study area is composed mainly of the remains of Sphagna (*Sphagnum fuscum, S. magellanicum*), Ericaceous shrubs and cottongrass (*Eriophorum vaginatum*) (Mathijssen et al. 2017). After drainage the remains of forest mosses and woody roots have increased their share in surface peat. Drainage has increased surface peat oxidation which is seen as a shallow layer of more decomposed peat about 10–20 cm below surface.

Remains of several forest fires are also present especially in the surface layers 30–50 cm, where a charcoal layer is clearly visible, but also in the deeper layers from 70 to 180 cm.

The mean air temperature in 2005–2008 was 15.3 °C during summer months (June–August) and –3.8 °C during winters (December–March). The annual mean air temperature was 5.1 °C and temperature sum (> 5 °C) 1356 d.d. Annual average precipitation was 722 mm and maximum wintertime snow depth 20–60 cm.

**2.2 Measurement setup**

The site was set up for C flux measurements in June-August 2004. The micrometeorological EC measurements were conducted from a mast, erected in August 2004, in the centre of the peatland at a 200–250 m distance to an upland forest in the north-west and to a small lake in the south-west. To the north-east the homogenous fetch was longer, about 600 m. The EC footprint was thus concentrated to the fairly homogenous peatland pine forest with at least 200 meter radius (Lohila et al. 2011).

The chamber measurements of $CO_2$ and $CH_4$ fluxes were conducted at four plots, located 50–100 m from the mast. The measurement collars were inserted and litter and plants removed from the treated collar plots in June 2004. As every plot consisted of 16 measurement points (collars), the whole setup contained 4×16, i.e. 64 measurement points. In addition, $CH_4$ fluxes from ditches were measured in 2011 at four points on two parallel ditches located on both sides of the mast.

The depth of the water table (WT) was manually measured from two perforated plastic pipes at each plot, along with chamber
measurements. WT was also continuously recorded close to the EC mast by a PDCR 830 (Druck Messtechnik GmbH, in 2004–2006), and a Hobo U20-001-01 (Onset Computer Corporation, MA, USA, in 2007–2009). Soil temperatures were recorded with temperature loggers (i-Button DS1921G, Maxim Integrated Products) from the depths of 5 cm ($T_5$) and 30 cm ($T_{30}$) below soil surface at intervals of 1–3 h. In 2005–2006, $T_5$ was recorded from every measurement point and $T_{30}$ from 16 points (four/plot). In 2007-2008, $T_5$ recordings were taken from two points/plot and $T_{30}$ recordings from two points in total.

The tree stand, ground vegetation and soil properties were measured on 33 plots located evenly along eight radial transects extending 160 m from the mast (the centre plot). Four transects with plots spaced at 20, 60, 100 and 140 m distances from the mast were alternated with four other transects with plots spaced at 40, 80, 120 and 160 m distances from the mast. The area of each plot was 200 $m^2$.

**2.3 Measurements**

**2.3.1 Ecosystem-atmosphere exchange of $CO_2$**

The turbulent fluxes of $CO_2$, water vapour ($H_2O$), sensible heat and momentum were measured with the eddy covariance technique on top of the 21.5-m telescopic mast (17.5 m from April 2005 to April 2006). Supporting meteorological

measurements included e.g. relative humidity (RH), photosynthetic photon flux density (PPFD), air and soil temperatures and soil moisture (see Lohila et al. 2011 for closer description). Measurements were carried out from August 2004 to March 2009. Here we report results for the full years 2005–2008.

We used an SATI-3SX (Applied Technologies, Inc.) sonic anemometer/thermometer from 2004 to November 2006, after which a METEK USA-1 (METEK GmbH, Elmshorn, Germany) was used. The atmospheric concentrations of $CO_2$ and $H_2O$ were measured with an LI-7000 (LI-COR, Inc.) analyzer. This instrument was calibrated bimonthly to monthly with two known $CO_2$ concentrations $[CO_2]$ (0 and 421 ppm). $CO_2$-free synthetic dry air was used as a reference gas. The heated inlet tube (3.1 mm Bevaline IV) for the LI-7000 was 17 m long, and a flow rate of $6\,l\,min^{-1}$ was used.

The signals were sampled at a frequency of 10 Hz, and the turbulent fluxes were calculated on-line as 30-min averages applying standard EC procedures. The effect of density fluctuations related to the water vapour flux (Webb et al. 1980) was included in the calculations, and the fluxes were corrected for systematic losses using the transfer function method of Moore (1986), including the losses due to autoregressive running mean filtering and the imperfect high-frequency response of the measurement system. Details of the flux calculation and correction procedures can be found in Pihlatie et al. (2010) and Lohila et al. (2011).

To estimate the storage fluxes of $CO_2$, the mean $[CO_2]$ observed at a height of 4 m with a LI-820 $CO_2$ analyzer and the $[CO_2]$ measured at the top of the mast were used. The storage term was calculated with the central difference method from the mean concentration during the subsequent and preceding 30 min periods and added to the measured turbulent flux. Hereafter NEE refers to the sum of turbulent and storage fluxes. In this paper, we use the convention that a positive value of NEE indicates a flux from the ecosystem to the atmosphere.

**2.3.2 Forest floor $CO_2$ efflux**

$CO_2$ efflux from forest floor was measured with an opaque closed steady state chamber (diameter 31.5 cm, height 14.9 cm) attached to a portable infrared gas analyzer (EGM-4, PP-Systems, Hitchin, U.K.; NSF11 in Pumpanen et al., 2004). Chamber closure time was 81 s. Measuring points were delimited with permanent collars and had four different treatments including the following respiration components: A) peat soil (including cut roots), B) A + above ground litter, C) B + living roots, and D) C + ground vegetation. In plots with vascular plants, extra collars of 5–10 cm height were used to fit the plants inside the chamber. The chamber volume was corrected accordingly.

In order to exclude autotrophic respiration, treatments A and B involved trenching with 30 cm deep collars and removing aboveground parts of living vegetation by repeated clippings every time before measurements if new plant growth had emerged. From treatment A, the above ground litter was also removed every time before measurements. From treatment C, only the above ground parts of plants were removed and treatment D was left intact. Collar depth in treatments C and D was only 2–3 cm to minimise disturbance to roots. Treatment D ($R_D$) thus includes all respiration components of forest floor respiration ($R_{FF}$) and treatment A respiration from peat soil only ($R_{PEAT}$). Respiration from treatment B ($R_B$) equals heterotrophic respiration ($R_{HET}$), and autotrophic respiration ($R_{AUT}$) is calculated as $R_D$-$R_B$. Autotrophic respiration of aboveground vegetation ($R_{GV}$) is defined as $R_D$–$R_C$, root respiration ($R_{ROOT}$) equals $R_C$–$R_B$ and $R_{LITTER}$ equals $R_B$–$R_A$.

$CO_2$ fluxes from treatments A and D were measured during the whole period 2005–2008, while treatments B and C only from 2005 to 2006.

### 2.3.3 Forest floor and ditch CH₄ fluxes

Soil $CH_4$ fluxes from the strips between ditches were measured with static chambers from the D points and reported by Lohila et al. (2011). To complement the $CH_4$ flux estimate for the whole area, fluxes from ditches were measured with the same equipment and methods as earlier. Fluxes were measured from four points on two parallel ditches on the both sides of the mast, altogether 7 times between June 28th and December 8th, 2011. The annual flux was estimated as 365 × daily mean flux.

### 2.3.4 Organic carbon pools and fluxes

The carbon stock in peat, and biomasses and litter production of the tree stand and ground vegetation, were measured to estimate organic carbon pools and fluxes in the peatland. Peat C stock was estimated based on average peat layer thickness on the tree stand transects (Lohila et al. 2011) and average carbon density in peat (Mathijssen et al. 2017). Tree stand properties were measured in spring 2005 and fall 2008. In 2005, the sample trees were cored to estimate diameter increment during the previous 5 years. Tree stand biomasses and C pools for years 2000, 2005 and 2008 were then estimated from these data using models of Repola (2008, 2009) and Laiho and Finér (1996) for pine belowground biomasses (root d>1 cm), as described in detail by Ojanen et al. (2012). In all biomass C stock and flux calculations, C content of 50% was assumed.

Above-ground biomass of ground vegetation vascular plants was sampled along the tree stand transects (n plots = 39), from an area of 0.25 m²/plot. Moss samples (n = 64) were collected from the same sites using corers with a diameter of 93 or 125 mm. In the lab, the dead part of the moss was cut and removed, based on ocular assessment (color change of the moss). The samples were separated by species and dry mass (105 °C) was determined for each sample.

The biomass of roots (and rhizomes of shrubs) were determined by taking a soil sample of 15 × 15 × 20 cm (width × length × depth) along the tree stand transects, adjacent to the mid points of the tree sample plots (n = 32). In the laboratory all roots were carefully separated from peat, divided according to species/functional groups (pine, spruce, birch, shrubs, grasses and herbs) and diameter (below and over 2 mm), dried in 105 °C and weighed. According to Bhuiyan et al. (2016), 15 % of the fine roots in Kalevansuo are located deeper than 20 cm. The biomasses estimated here were corrected accordingly.

C flux in above ground litter was estimated with 14 litter traps (20 × 20 cm) per chamber plot (i.e. altogether 56 traps). Litter was collected 2–3 times per year, separated by species, dried in 105 °C and weighed. As moss litter is not captured by litter traps, moss litter production was estimated by harvesting moss biomass production over 2 and 5 years (Ojanen et al. 2012). As the whole moss biomass eventually dies and forms litter on site, annual moss biomass growth equals annual litter production.

Coarse root (>2 mm) litter production was estimated as biomass × turnover rate (0.12 for pine, 0.08 for shrub rhizomes; Finér and Laine 1998). Fine root litter production was estimated with root-ingrowth-cores by Bhuiyan et al. (2016). Sixty cores (diameter 3 cm, length 50 cm) filled with *Sphagnum* peat were installed into soil in October 2009, and 20 cores were collected every year for three years. The fine root production rate was calculated as the average fine root mass (live+dead) in the cores divided by incubation years (average for 2nd and 3rd years).

### 2.3.5 Change in peat layer thickness

To survey the changes in peat layer thickness, caused by compaction and decomposition of soil organic matter, litter production and moss height growth, soil surface around the mast was levelled in 2004, 2011 and 2014. In the beginning of measurements in 2004, a 20 mm thick steel rod was hammered through the peat layer firmly to the subsoil, serving as a stable benchmark. The soil surface at the undisturbed chamber measurement points (D-collars), was repeatedly levelled in relation to the benchmark. A manual levelling instrument with a levelling rod was used and the readings were recorded with the precision of ±0.5 cm.

## 2.4 Gas flux calculations

### 2.4.1 NEE

The NEE data obtained from the EC measurements were screened as described by Lohila et al. (2011). In short, screening criteria were applied to remove spikes in the 10-Hz anemometer data and to discard poor-quality 30-min data. For the latter, the criteria were based on the expected range of the mean $[CO_2]$ and air temperature (from the sonic anemometer), and of the variances of $[CO_2]$, vertical wind speed and air temperature. In addition, a cumulative flux footprint of 70% was required, and a threshold of 0.1 m s$^{-1}$ was set to the friction velocity (Lohila et al. 2011). The procedures of gap-filling of the EC flux data and partitioning of NEE to the GPP and $R_{ECO}$ components are described in Appendix A. The estimation of uncertainties in annual NEE is described in Appendix B.

### 2.4.2 Forest floor $CO_2$ efflux

$CO_2$ efflux from forest floor is a result of heterotrophic and autotrophic processes from different layers (vegetation and soil), which have different temperature dynamics. Therefore an additive, layerwise model was used, in which soil temperatures $T_5$ and $T_{30}$ predict fluxes from different layers, with different temperature dynamics. An Arrhenius type function (Lloyd & Taylor 1994), was fitted to the measured $CO_2$ efflux (g $CO_2$ m$^{-2}$ h$^{-1}$) from forest floor:

$$CO_2\ efflux = R_{\text{REF5}} \exp\left[E_{05}\left(\frac{1}{T_{\text{REF}}-T_0} - \frac{1}{T_5-T_0}\right)\right] + R_{\text{REF30}} \exp\left[E_{030}\left(\frac{1}{T_{\text{REF}}-T_0} - \frac{1}{T_{30}-T_0}\right)\right] \tag{1}$$

where $R_{REF5}$ and $R_{REF30}$ are respirations at reference temperatures ($T_{REF} = 10$ °C) and $E_{05}$ and $E_{030}$ describe temperature sensitivities of respiration in 5 cm and 30 cm peat depths, respectively. $T_0 = -46.02$ °C is a constant.

Parameter values were estimated separately for different treatments (A–D) representing different components of $R_{FF}$, the four gas measurement plots, and two groups of years (2005–2006 and 2007–2008; Appendix C), as the decomposability of soil organic matter changes in time at A collars. WT was also tested as an explanatory variable, but as it predicted the temporal flux variation poorly, it was not included in the final models. The models were used with measured soil temperature data to simulate the temporal dynamics and annual fluxes of different flux components.

## 2.5 Modeling of the tree stand $CO_2$ fluxes

To analyse the contribution of the tree stand (aboveground) to the ecosystem $CO_2$ exchange, we used the GPP and shoot respiration (R) models in Stand Photosynthesis Program (SPP). SPP predicts canopy light interception, photosynthesis and shoot respiration in half-hourly time steps (Mäkelä et al., 2006). PPFD, air $CO_2$ concentration, air temperature, and relative air humidity measured at the site were used as inputs for SPP. The photosynthesis model used was OPAC (Mäkelä et al. 2006). Tree stand was described as three size classes (Ojanen et al. 2012), foliar masses for each class were estimated using the models of Repola (2009), and these were converted to leaf area index with specific leaf area of 11 m$^2$ kg$^{-1}$ (Luoma, 1997). Stem respiration was estimated with the model of Zha et al. (2004).

## 3. Results

### 3.1 Meteorological conditions

Of the studied years, 2008 was the warmest, especially during the winter months January–March, which were almost snowless. It was also the rainiest year. The summer (June to August) of 2008 was significantly cooler, but otherwise similar to the other

summers. In contrast, the year 2006 was exceptionally dry from January until the end of September, including a severe drought during the growing season. In summer 2006, air temperature and PPFD were higher than on other years, whereas relative humidity and water table were lower (Table 1, Figs. 1 and 2). The dry and warm growing season 2006 was preceded by a cold winter, which is why soil surface temperatures ($T_5$) were much below average in the spring, and down in 30 cm stayed below average until September, i.e. for almost the whole growing season (Fig. 2). In September–October the deeper peat layers finally warmed up and stayed warmer than average for the rest of the year.

WT typically fluctuated between –30 and –50 cm in a year, being on average 42 cm below ground surface during the snow free season (April–November) and only about 5 cm higher during the winters (December–March) (Figs. 1 and 2). WT varied also spatially (mean range between water-wells 24 cm), being deeper in the hummocks (–49 cm) compared to the lawns (–35 cm). During the drought in 2006, WT started dropping down in July, reached –79 cm in the end of September, and rose again after heavy rainfalls in the beginning of October. The average WT in 2006 was 10 cm deeper than in other years.

## 3.2 Ecosystem CO₂ exchange

According to the EC flux measurements, the site acted as a $CO_2$ source typically during the winter months (October–March) and a sink during the growing season (April–September) (Figs. 3, 4a). The variation in NEE during winter was small, ranging from about –0.1 to 0.1 mg m$^{-2}$ s$^{-1}$ (Fig. 3). While there were occasional, warm days with net $CO_2$ uptake during the winter, the actual spring recovery of photosynthesis seemed to occur typically in the beginning of April, the only exception being the spring after the warm winter of 2006–2007, when the recovery started already in March. In summer (June–August), the highest night-time $CO_2$ emission values, representing $R_{ECO}$, were on average 0.35 mg m$^{-2}$ s$^{-1}$, and the highest day-time $CO_2$ uptake typically fluctuated around –0.75 mg m$^{-2}$ s$^{-1}$. Only in summer 2006 the amplitude in the diurnal dynamics was smaller.

The site was a sink of $CO_2$ in all years, NEE varying between –520 and –990 g $CO_2$ m$^{-2}$ a$^{-1}$ (Table 2). The average NEE for the four years was –860 g $CO_2$ (i.e. –234 g C m$^{-2}$ a$^{-1}$). With the exception of the dry year of 2006, the annual NEE was surprisingly similar in other years, varying from –950 to –990 g $CO_2$ m$^{-2}$ a$^{-1}$. The estimated uncertainty in the annual budget, including the random measurement and gap-filling error and the uncertainty in the high-frequency loss correction and gap-filling of the longer than 2-day gaps, varied from 35 to 114 g m$^{-2}$ a$^{-1}$, corresponding to 3.6 to 22% of the respective annual balance (Appendix B).

The drought during the spring and the growing season of 2006 was clearly reflected in the $CO_2$ exchange. The (gap-filled) NEE and GPP were markedly less negative in June and July 2006, indicating lower $CO_2$ uptake by photosynthesis as compared to the other years (Figs. 4b, c). However, in July and August $R_{ECO}$ was also clearly suppressed (Fig. 4d), thus decreasing the net loss of $CO_2$ from the peatland (NEE). In October 2006, GPP had fully recovered to the level of other years, but $R_{ECO}$ stayed at a slightly higher level during the rest of the year, leading to clearly higher NEE during the last months of the year.

After the first week of June until the end of July 2006, there were only a few days with accepted NEE observations (Fig. 3), so the results shown for these months (Fig. 4) largely depend on gap-filling. However, the main parameters of respiration and photosynthesis ($R_{REF}$ = respiration at 10 °C, $GP_{MAX}$ = photosynthesis in optimal light conditions; see Appendix A) indicate that both the photosynthetic capacity and ecosystem respiration were reduced in the summer of 2006 (Fig. 5). The data coverage was considerably better in August, making it possible to reliably study the impact of drought on NEE. In August 2006, both $R_{REF}$ and $GP_{MAX}$ had values that were significantly different from the other years (Fig. 5). While typically $R_{REF}$ reached its maximum in August and decreased thereafter ($GP_{MAX}$ having similar but opposite dynamics), in 2006 the trend was reversed and both $R_{REF}$ and $GP_{MAX}$ increased towards October. This suggests that the ecosystem was affected by the drought in August and September 2006 and slowly recovering in October. Thus, the distinct decrease in the annual net $CO_2$ uptake in 2006 (Table

2) was likely to be caused by the GPP decrease during the summertime, although $R_{ECO}$ decreased during the drought as well. In addition to the summer depression in net $CO_2$ uptake, the higher $R_{ECO}$ in autumn months after the drought and heavy rains in October (Fig. 2) furthermore increased the difference to other years: the cumulative NEE in October-December in 2006 was as high as 320 g $CO_2$ m$^{-2}$, while in other years it varied from 130 to 190 g $CO_2$ m$^{-2}$.

## 3.3 Forest floor CO$_2$ flux

The measured, instantaneous $CO_2$ fluxes from forest floor ($R_{FF}$) varied between –0.02 and 1.80 g $CO_2$ m$^{-2}$ h$^{-1}$ (Fig. 6), following the dynamics in soil temperature. For the treatments A, B, C and D, the mean ± s.d. respiration fluxes in non-winter seasons (April-November) 2005–2006 were 0.23±0.11, 0.31±0.13, 0.38±0.22 and 0.42±0.24 g $CO_2$ m$^{-2}$ h$^{-1}$, respectively. In winter (i.e. over a snowpack or frozen ground between December and March), the mean fluxes were almost the same in the different treatments (0.022, 0.019, 0.022 and 0.035 g $CO_2$ m$^{-2}$ h$^{-1}$ from A to D, respectively).

The regression models with $T_5$ and $T_{30}$ as explanatory variables (Eq. 2) explained 70% (46%–90%) of the variation in the fluxes of the entire dataset (Appendix C). Respiration rates at 10 °C ($R_{REF5}$ and $R_{REF30}$) increased from A to D collars, i.e. as respiration components were added, and decreased at A collars with time since the beginning of the study (05–06 to 07–08).

The modelled annual respiration ranged during the first two years from 1233 g $CO_2$ m$^{-2}$ a$^{-1}$ in A collars ($R_{PEAT}$) to 2312 g $CO_2$ m$^{-2}$ a$^{-1}$ in D collars ($R_{FF}$, Table 3). During 2007–2008, $R_{PEAT}$ clearly decreased from the previous years, to ca. 830 g $CO_2$ m$^{-2}$ a$^{-1}$, whereas $R_{FF}$ varied little between the studied years. In A collars the decomposability of organic matter is likely gradually decreased when the labile components are decomposed and the recalcitrant ones are enriched. Also, as we had to remove a newly grown moss layer from A-collars in in spring 2007 (inevitably with some soil organic matter attached), this procedure probably decreased the proportion of labile components on the soil surface.

Based on the modelled fluxes of the first two years, $R_{HET}$ contributed 75% and $R_{AUT}$ 25% to the mean annual $R_{FF}$ (Table 3). $R_{PEAT}$ comprised 53% of the flux, $R_{LITTER}$ 22%, $R_{ROOT}$ 16% and $R_{GV}$ 8%. The four-year mean of $R_{FF}$ was 2197 g $CO_2$ m$^{-2}$ a$^{-1}$, i.e. ca 600 g C m$^{-2}$ a$^{-1}$. Using this mean value with the proportions from 2005–2006, we get an estimate for $R_{HET}$ of 450 g C m$^{-2}$ a$^{-1}$ and $R_{AUT}$ 150 g C m$^{-2}$ a$^{-1}$.

In 2006, the main part of summertime (June 15th to September 12th) measurements were lost due to instrument failure. Thus, we cannot reliably analyse the impact of 2006 summer drought on forest floor respiration. The existing soil $CO_2$ efflux data from September 2006, when WT was extremely low, do show higher effluxes than those in early June 2006, although soil surface temperatures ($T_5$) were lower in September. However, at the same time $T_{30}$ was much higher (10.7 °C) than in June (5.4 °C), explaining the increased efflux. Compared to the other years, soil temperatures in September were at their highest in 2006 (Fig. 2), and the temperature response models thus predicted higher fluxes for September 2006 than for the other years. Following the heavy rains in the beginning of October, respiration decreased at the same time with the the rise of WT – and the decrease in $T_5$.

The impact of WT on forest floor respiration was ambiguous. Correlations between WT and $CO_2$ efflux were weak and variable by year and treatment. The residuals of the model (Eq. 1) estimates vs. WT indicated a positive response especially in D collars (lower $R_{FF}$ with lower WT). However, this effect was caused mostly by spatial variation, as measurement points in hummocks generally had lower WT and lower respiration than the points in the lawn-level. Since the models were used for predicting temporal dynamics, WT was not included in the models.

### 3.4 Simulated tree stand CO₂ flux

The SPP-model simulated the tree stand GPP and respiration well. For the year 2008 with the most complete NEE data, the $R_{ECO}$, derived from the gap-filling and partitioning of the EC measurements, matched very well (0.8% difference) the model-derived sum of $R_{FF}$ and above-ground tree respiration ($R_{TREE}$, Fig. 7a, Table 2). Not surprisingly, the model was not able to simulate the suppression of respiration in 2006 (Fig. 7b), apparently since it does not have linkages to soil moisture. The simulated four year average was 9% higher than the EC-derived $R_{ECO}$ (Table 2).

The simulated four-year average GPP of the tree stand was 2473 g $CO_2$ m$^{-2}$ a$^{-1}$ (675 g C). The GPP for the ground vegetation, measured by manual flux chambers in another campaign, was 1040 g $CO_2$ m$^{-2}$ a$^{-1}$ (Badorek et al. 2011). Altogether the tree stand and the ground vegetation GPP sum up to 3513 g $CO_2$ m$^{-2}$ a$^{-1}$, which is relatively close (92%) to the ecosystem GPP obtained from the partitioning of the EC fluxes (3805 g $CO_2$ m$^{-2}$ a$^{-1}$). These independent findings suggest that the tree stand contributes about 70% and the ground vegetation 30% of the GPP at Kalevansuo.

### 3.5 CH₄ fluxes

CH₄ flux from ditches was very variable, especially spatially but also temporally. The instantaneous fluxes varied between –0.098 and 1.757 mg $CH_4$ m$^{-2}$ h$^{-1}$. The wettest plot, with cottongrass (*Eriophorum vaginatum*) emitted on average 0.936 mg $CH_4$ m$^{-2}$ h$^{-1}$, significantly more (p <0.001) than the other three, slightly drier plots, (mainly *Sphagnum riparium*), with mean fluxes of 0.006, 0.056 and –0.006 mg m$^{-2}$ h$^{-1}$. Temporal variation was high but no clear seasonality was observed. At the wettest plot, fluxes had similar temporal pattern with WT, i.e. the highest flux took place in September during the highest WT.

The average flux from ditch plots was 0.248 mg $CH_4$ m$^{-2}$ h$^{-1}$, which calculated for the whole drained area (ditches 2.5% of the area) increased the estimated total flux by 0.006 mg $CH_4$ m$^{-2}$ h$^{-1}$. As the flux at the strips was on average –0.015 mg m$^{-2}$ h$^{-1}$ (Lohila et al. 2011), the site would therefore remain as a small sink for CH₄. The annual areally weighted flux was –0.06 g $CH_4$ m$^{-2}$ a$^{-1}$ (i.e. –0.12 g $CH_4$ m$^{-2}$ a$^{-1}$ × 0.975 (strips) + 2.2 g $CH_4$ m$^{-2}$ a$^{-1}$ × 0.025 (ditches)).

### 3.6 Change in peat layer thickness

The soil surface on the undisturbed D collars had subsided on average by 1.4 cm in ten years from 2004 to 2014, i.e. 1.4 mm a$^{-1}$ (Fig. 8). There was considerable variability between points from an increase in elevation by 2 cm to a subsidence of 5 cm, so that the change was not quite statistically significant (p=0.067). Also, some back and forth variation in peat thickness between years was observed: in August 2011 all but four points had lower elevation than in 2014. This can be either a measurement error or real shrink-swell behaviour (breathing) of the peatland.

### 3.7 Carbon balance

The biggest carbon pool at Kalevansuo (Fig. 9.) was the 2.2 m thick peat layer making up 95.3% of the total carbon pool. Tree stand (without fine roots) comprised 4.3% and ground vegetation only 0.4%. Fine roots comprised 0.2%. The total C pool in vegetation in 2008 was 5.5 kg m$^{-2}$, which corresponds to about 10 cm layer of peat. Aboveground parts comprised 62% of the total biomass. Of the moss biomass, *Sphagna* comprised 20% and forest mosses 80%.

The tree stand volume increased from 90 m³ ha$^{-1}$ in 2000 to 130 m³ ha$^{-1}$ in 2008, i.e. on average by 5 m³ ha$^{-1}$ a$^{-1}$. The corresponding carbon pool was 4.6 kg m$^{-2}$ in 2008 and 3.2 kg m$^{-2}$ in 2000. The tree stand thus sequestered ca. 170 g C m$^{-2}$ a$^{-1}$. This made 74% of the carbon accumulation at Kalevansuo, while the rest was attributed to peat soil (Fig. 9).

Total litter production was estimated at 437 g C m$^{-2}$ a$^{-1}$. Of this mosses comprised 20% and vascular plants 80%. Of the litter production by vascular plants, trees comprised 79% (aboveground) and 66% (belowground). Fine root production was estimated at 120 g C m$^{-2}$ a$^{-1}$ (Bhuiyan et al. 2016), comprising 76% of the belowground litter.

As the average of the four years, the Kalevansuo peatland ecosystem fixed ca. 1040 g C m$^{-2}$ a$^{-1}$ through photosynthesis, 70% of which was attributed to the tree stand. Simultaneously it lost 810 g C m$^{-2}$ a$^{-1}$ through R$_{ECO}$. Ca. 50% of R$_{ECO}$ resulted from heterotrophic respiration and 50% from autotrophic respiration of trees and ground vegetation. R$_{FF}$ was comprised mainly of heterotrophic respiration of peat and litter (75%), and less by autotrophic respiration of tree roots and ground vegetation (25%).

Some C may have been lost through leaching (not measured), but this is considered a minor component due to ineffective ditches and high transpiration. No C was lost as methane, as the site was a small CH$_4$ sink (–0.06 g CH$_4$ m$^{-2}$ a$^{-1}$), which is insignificant for the C balance.

## 4. Discussion

### 4.1 Ecosystem CO$_2$ fluxes — the effects of drought

The Kalevansuo drained peatland forest was a strong CO$_2$ sink in all the four years studied (2005–2008). The annual sinks were similar, except for the dry year 2006, when it was only about 50% of that in other years. Interestingly, this decrease in ecosystem CO$_2$ sink was not caused by increased R$_{ECO}$ in drier conditions, as could be expected. Both GPP and R$_{ECO}$ were reduced in summer, and the reduction in GPP was larger. In addition, the higher-than-normal soil temperatures in September and October and the very high precipitation in October resulted in higher R$_{ECO}$ in autumn 2006, which partly explained the much lower annual CO$_2$ net uptake.

Despite the long gap in the NEE data in June and July 2006, we were able to demonstrate with the data from August, one of the driest months, that drought had a clear impact on the potential CO$_2$ exchange. Based on the direct responses between the night-time NEE (respiration) and temperature, and between the daytime NEE and PPFD, the parameters describing the potential ecosystem respiration and daytime net CO$_2$ uptake were reduced in August 2006 as compared to the other years. However, GPP was not markedly different from the other years due to the larger number of clear-sky days with higher irradiation levels. On the other hand, the higher temperatures were not able to compensate for the reduced respiration potential (i.e. the parameter R$_{REF}$), resulting in a reduced monthly R$_{ECO}$.

Drought has been shown to strongly affect NEE through decreased GPP in pristine mires where vegetation is adapted to high water table (Alm et al. 1999, Bubier et al. 2003, Lafleur et al. 2003). Although Scots pine, the main tree species in Kalevansuo peatland, is a drought-tolerant species, summer droughts have been reported to decrease its radial growth in drained peatlands (Huikari and Paarlahti 1967). The water table in Kalevansuo is usually rather high, which means that the roots of pines are located mainly in the top 40 cm (Bhuiyan et al. 2016), i.e. in the oxic layer above the average water table. During drought, when water table may drop down to 80 cm for several weeks, even the pines will probably suffer from water deficit, and close their stomata.

In contrast to GPP, R$_{ECO}$ and soil respiration have often been shown to increase in peatlands, when water table is lowered and more peat is exposed to oxidation (e.g. Silvola et al. 1996, Flanagan and Syed 2011, Ballantyne et al. 2014, Munir et al. 2014, 2017). However, many studies have shown only a weak or no impact of WT on R$_{ECO}$, whereas soil temperature has been driving the respiration fluxes (Lafleur et al. 2005, Nieveen et al. 2005, Juszczak et al. 2013, Olefeldt et al. 2017). In Kalevansuo, the latter seems to be the case. R$_{ECO}$ was slightly lower during the drought in August 2006 compared to other years (Fig. 5).

$R_{FF}$ was strongly controlled by soil temperatures, whereas WT had only a weak and varying effect in different treatments and years.

The decrease in $R_{ECO}$ may be caused by decrease of both $R_{AUT}$ and $R_{HET}$. As the drought decreases GPP, it will decrease also photosynthetically driven autotrophic respiration (Olefeldt et al. 2017), while heterotrophic respiration may well continue in deeper, still moist but now more oxic, peat layers. However, a large part of $R_{HET}$ is originated from the decomposition of the new organic matter (Chimner and Cooper 2003), i.e. above ground and root litter, deposited mainly in the very surface of the peat soil. In drained peatlands the decomposition rate of this surface layer is hardly ever restricted by too high WT, but sometimes it can be restricted by too low moisture content (Mäkiranta et al. 2009).

If water levels were lowered for a longer period, e.g. through deeper ditching, the effect might be different than that of drought: a more efficient drainage would induce higher decomposition and heterotrophic respiration through changes in microbial communities (Mäkiranta et al. 2009) but also probably increased root growth into the deeper layers.

### 4.2 Soil subsidence

Even though the flux and biomass data indicate a steady increase in soil C stock, a small (insignificant) subsidence of the soil surface was measured (0.14 cm/year). The value is considerably smaller than that reported for agricultural fields (0.3–3 cm/year, Oleszczuk et al. 2008), or for palm oil plantations on peat with high observed C losses (4.2 cm/year; Couwenberg and Hooijer 2013). In peatlands drained for forestry, subsidence is in long-term usually much smaller (Lukkala 1949, Minkkinen and Laine 1998a) because of shallower drainage and continuous litterfall and humus formation on the soil surface. The only published long-term study from drained peatland forest reports rates of 0.4–0.7 cm/year for a dwarf-shrub site in southern Finland (Ahti 2002).

Subsidence of peat is caused by physical compaction and loss of organic matter through oxidation. In physical compaction, solid matter is compacted into a smaller space. The result is the increase in bulk density, which is evident in all drained peatlands (e.g. Minkkinen and Laine 1998b). We do not have bulk density measurements from Kalevansuo peatland prior to drainage, but compared to similar pristine sites (38 kg m$^{-3}$ natural pine mires (Minkkinen and Laine 1998), bulk density of the surface 0-20 cm layer is higher in Kalevansuo (94 kg m$^{-3}$). It is therefore likely that bulk density has increased in Kalevansuo after drainage. In oxidation, organic matter is lost as $CO_2$ from the peat to the atmosphere. In peat soil, both processes take place at the same time, and in forested sites especially, the C loss through oxidation is to varying extent compensated for by litter production. Thus, given the estimated positive soil C balance (i.e., accumulation of C in the soil) at Kalevansuo, we conclude that the observed small subsidence is caused by compaction, not by loss of peat.

### 4.3 Carbon balance

Kalevansuo accumulated atmospheric C every year studied. Given that the average net carbon uptake of the site was 230 g m$^{-2}$ a$^{-1}$ and that 170 g m$^{-2}$ a$^{-1}$ was sequestered to the growing tree stand, the remaining 60 g C m$^{-2}$ a$^{-1}$ must have been accumulated in the other parts of the ecosystem. If the ground vegetation biomass is assumed constant, the surplus must be in the peat soil. This assumption is based on ocular assessment at the site. It is reasonable to assume that the ground vegetation biomass is not increasing, since the tree stand is steadily growing bigger and the correlation between tree stand and ground vegetation biomass is negative (Reinikainen et al. 1984). Furthermore, an increase of 60 g C m$^{-2}$ a$^{-1}$ would equal the doubling of shrub biomass in 5 years, and that should be clearly visible. Thus the method should not be overestimating soil C pool increase. However, as the C pool in ground vegetation is one-tenth of that in the tree stand, the change in C pool would be irrelevant, assuming the same relative growth rate.

Despite the small biomass pool compared to the tree stand, ground vegetation was estimated to produce above ground litter at a rate of 130 g C $m^{-2}$ $a^{-1}$, i.e. almost as much as the tree stand (Fig. 9). The majority of this litter originates from mosses, the coverage of which is almost 100% in Kalevansuo. Another rapidly renewing biomass pool was that of fine roots, which was composed almost totally of tree and shrub roots. About half of this pool is renewed annually, producing root litter at a rate of 120 g C $m^{-2}$ $a^{-1}$ (Bhuiyan et al. 2016). When decomposed, part of the released C is translocated as a solute into deeper peat layers (Domisch et al. 2000). Thus, although being small C pools, both ground vegetation and fine roots have a large impact on the soil C balance.

In our estimation, the C in the below ground parts of trees (stumps and roots > 1 cm diameter) was considered as tree biomass, which increases as the stand grows. When trees die, either naturally or as they are harvested, the below-ground part of C becomes a part of the soil C pool. Considering this below-ground biomass as a part of the soil C pool, would increase the soil C accumulation estimate to over 100 g C $m^{-2}$ $a^{-1}$. The biomass of smaller roots could of course also change, but as the biomass pool of the 2–10 mm roots is only a small fraction of that of the bigger ones (Fig. 9), and as the fine root turnover is rapid (50% $a^{-1}$), this is not considered a major uncertainty.

Taking into account the leaching of C would have only a minor effect on the NEE estimate. We do not have dissolved organic carbon (DOC) measurements from Kalevansuo, but leaching of DOC, i.e. the output of dissolved C, from Finnish drained peatlands ranges from 10 to 15 g C $m^{-2}$ $a^{-1}$ (Sallantaus and Kaipainen, 1996; Kortelainen et al., 1997; Sarkkola et al., 2009; Rantakari et al., 2010). This is 4–7% of the estimated NEE and 17–25% of the soil C balance at Kalevansuo. As the ditches in Kalevansuo are ineffective and the transpiration of the tree stand and ground vegetation is an important pathway for water output (Sarkkola et al. 2010), leaching of DOC at Kalevansuo is likely at the lower end of the observed range. Thus, taking leaching into account would not change the conclusion on soil C sink.

Based on four-year NEE and tree growth data, we estimated that the accumulation of C in soil was on average 60 g C $m^{-2}$ $a^{-1}$ during the four-year period. Since the tree stand growth data is based on five-year average, we cannot say whether the soil C balance has been positive in all the studied years. Neither can we say if the long-term soil C balance of the peatland would stay similar in the future. In natural mires, where long-term peat C accumulation can be reliably estimated from peat coring and radiocarbon ($^{14}$C) dating, the multi-year mean NEE derived from EC measurements has typically been similar to the long-term accumulation rate (Aurela et al. 2004, Roulet et al. 2007, Nilsson et al. 2008), although not in all cases (Ratcliffe et al. 2017). It is well-known that long-term average rates, determined by peat coring and radiocarbon dating, are not necessarily the same as the actual, current (or decadal average) rates (e.g. Clymo et al. 1998, Frolking et al. 2014).

Kalevansuo has been cored extensively, and historical C accumulation has been determined using radiocarbon dating (Mathijssen et al. 2017). However, the drainage took place so recently (35 years before our study) that post-drainage C accumulation cannot be reliably determined using $^{14}$C dating. Even if the surface peat could be dated accurately, root growth into deeper layers would mess up the C accumulation estimate. Several peat-coring methods have been tried to estimate post-drainage changes in peat C stocks (e.g. Kruger et al. 2016, Minkkinen and Laine 1998, Minkkinen et al. 1999, Simola et al. 2012, Turetsky et al. 2004) but they all have large uncertainties. However, as discussed above, we were not trying to estimate long-term peat accumulation, only the current rate. Eddy covariance combined with biomass growth measurements is the most accurate method for this purpose.

Ojanen et al. (2012) evaluated different chamber-based methods for calculating the soil C balance, and compared these to the EC-based method described above. The "L–$R_{HET}$ -method" (litter production minus heterotrophic respiration) produced varying results depending on the variable fine root turnover rates available from literature. Using the recent results of fine root production in Kalevansuo (Bhuiyan et al. 2016) we end up with L of 437 g C $m^{-2}$ $a^{-1}$ – and $R_{HET}$ of 450 g, which results in a

loss of 13 g C m$^{-2}$ a$^{-1}$. Thus there is still a difference of about 73 g C m$^{-2}$ a$^{-1}$ to the EC-based estimate. This difference is probably caused by uncertainties in estimating $R_{HET}$ (Ojanen et al. 2012). The cutting of roots causes an extra litter input (e.g. Subke et al. 2006) and on the other hand prevents further input. Roots may also reach under the 30-cm deep collars (Bhuyian et al. 2016). Trenching also affects soil moisture that regulates respiration (Subke et al. 2006).

In addition to the "L–$R_{HET}$ -method", soil C balance can be estimated using data from transparent chamber and tree litter measurements, as follows:

Soil C balance = GPP$_{FF}$ – L$_{TREE}$ + R$_{FF}$ – R$_{AUT}$ of tree roots          (2)

where GPP$_{FF}$ is chamber measured GPP of forest floor vegetation and L$_{TREE}$ total litter from trees. Since the chamber-measured R$_{FF}$ includes also tree stand root respiration this must be subtracted from R$_{FF}$.

We estimated GPP$_{FF}$ at –288 (Badorek et al. 2008), R$_{FF}$ at 600, R$_{AUT}$ of tree roots at 89 and L$_{TREE}$ at 253 g C m$^{-2}$ a$^{-1}$ (Fig. 9). This gives an estimate for the soil C balance of –30 g C m$^{-2}$ a$^{-1}$ (sink), which is relatively close to the EC-based estimate of – 60 g C m$^{-2}$ a$^{-1}$, and supports our finding of the soil C sink.

## 4.4 Can the carbon sink last?

Here we have shown that the Kalevansuo drained peatland ecosystem and even the soil is currently a carbon sink despite the
drainage. It would be reasonable to assume that drainage would turn a peatland soil into a carbon source, because the decomposition of peat is typically increased after drainage. Drainage in Kalevansuo is, however, rather superficial, the average water table being at 35–40 cm, i.e. only about 15–20 cm lower than in natural dwarf-shrub pine bogs (Minkkinen et al. 1999). The site is topographically rather even, as is typical for nutrient-poor pine bogs, so draining with open ditches is not efficient. Thus the ditches are blocked by vegetation and drainage is mainly mediated by the transpiration through the tree stand
(Sarkkola et al. 2010). The soil is also almost fully covered with vegetation, including mire species like *Sphagnum* mosses. Such a small change in vegetation structure is typical for drained dwarf shrub pine bogs (Minkkinen et al. 1999). It thus appears that this peatland has not lost the ability to keep up the relatively high water table and surface moisture supporting the continuous growth of mosses. Only very dry seasons, like summer 2006, may disturb the hydrology so much that C dynamics are seriously affected.

It is evident that most boreal and temperate peatland forest ecosystems, where drainage has been successful, act as contemporary C sinks (Ojanen et al. 2013, Meyer et al. 2013, Hommeltenberg et al. 2014), because the tree stand C sequestration exceeds the loss of C from soil. In peatlands used for forestry it is however the soil C storage that is important in the long-term, given that the tree stock will eventually be harvested and the C in wood products will gradually be lost back to the atmosphere. Thus the most relevant question is: Will sites like Kalevansuo remain C sinks in the long-term if they are
managed for forestry? After the site is harvested, as typical, by clear-cutting, soil decomposition processes will go on, whereas litter production from tree stand is ceased for several years. Logging residues will decompose rather fast, and may enhance the decomposition rate of the underlying peat soil (Mäkiranta et al. 2012, Ojanen et al. 2017). This will create a loss of soil C through soil respiration, the magnitude of which is dependent on soil quality (von Arnold et al. 2005a, b, Minkkinen et al. 2007).

On the other hand, in typical stem-harvesting method, tree stumps and roots are left at the site, increasing the C stock in the soil significantly. This C pool of coarse woody debris is not easily decomposed (Laiho and Prescott 2004) especially when buried in peat soil, and its inclusion will compensate for the soil C losses for several years. Also, after clear-cut, the water table will rise because of the removal of the transpiring tree stand, likely reducing peat decomposition rate (Mäkiranta et al. 2010).

This reduction is, however, probably quite small and the site is likely to be a strong C source at least for the first five years, after which the growing vegetation again starts to bind carbon to the system (Mäkiranta et al. 2010, Kolari et al. 2004). However, no data of C dynamics of the young stand phase on forested peatlands exist. To answer the question of the climatically best option to manage different kinds of drained peatlands, simulations with mechanistic models verified for peatland conditions (e.g. He et al. 2016) are promising tools.

## 5. Conclusions

Despite the drainage, the Kalevansuo peatland forest in southern Finland was a strong carbon dioxide sink during all the four years studied. The peat soil also accumulated carbon at an estimated mean rate of 60 g C $m^{-2}$ $a^{-1}$. Kalevansuo was thus a similar C sink to natural peatlands in general. In addition, the site was a small $CH_4$ sink, in contrast to natural mires. Based on earlier knowledge of similar sites on drained peatlands, Kalevansuo is not an exception, but rather represents a typical drained pine bog, regarding the greenhouse gas fluxes. Modelling studies, in addition to further measurements focusing on young stands the first 20 years after cuttings would be necessary to show whether the sink is maintained under long-term production forestry.

Drought affected the potential and actual $CO_2$ fluxes and had a strong impact on the C balance of Kalevansuo mainly through the decrease in photosynthesis, although the simultaneously suppressed respiration decreased the potential C loss from the system. On the other hand, the high late-autumn $CO_2$ emissions occurring after the heavy rains in October partly explained the smaller annual net $CO_2$ uptake of that year. However, the site remained a clear $CO_2$ sink even during the drought, suggesting that occasional droughts do not threaten the sink capacity of such peatlands.

## Acknowledgements

We thank Tiina Badorek for biomass measurements, Markku Koskinen for measuring ditch $CH_4$ emissions, Tea Thum helping in flux measurements and Timo Haikarainen and Inkeri Suopanki for help in tree stand measurements and computations. Funding for setting up the site was received from Ministry of Agriculture and Forestry and Ministry of Trade and Industry of Finland. Kolli foundation (Marjatta ja Eino Kollin Säätiö) funded the biomass measurements.

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

.

Table 1. Meteorological parameters for the full years and summer months, June-August. T = mean air temperature, P= precipitation sum, PPFD = mean daily sum of photosynthetic photon flux density, RH = mean relative humidity, VPD = mean vapour pressure deficit in the afternoon, 12:00–16:00 local time.

| Year | Year | | June–August | | | | |
| | T | P | T | P | PPFD | RH | VPD |
| | (°C) | (mm) | (°C) | (mm) | (mol m$^{-2}$ d$^{-1}$) | (%) | (kPa) |
| --- | --- | --- | --- | --- | --- | --- | --- |
| 2005 | 4.7 | 725 | 15.1 | 285 | 35.0 | 76.0 | 0.89 |
| 2006 | 5.1 | 600 | 16.5 | 95 | 37.6 | 66.9 | 1.35 |
| 2007 | 4.9 | 724 | 15.1 | 234 | 35.1 | 75.4 | 0.93 |
| 2008 | 5.6 | 839 | 14.3 | 237 | 31.3 | 73.2 | 0.91 |

Table 2. EC-measured (and gap-filled and partitioned) annual net ecosystem exchange (NEE ± error; Appendix B), gross primary production (GPP) and ecosystem respiration ($R_{ECO}$) of the Kalevansuo peatland, in comparison with the simulated tree stand GPP ($GPP_{TREES}$), tree stand above ground respiration ($R_{TREES\_AG}$) and forest floor respiration ($R_{FF}$). Unit: g $CO_2$ m$^{-2}$ a$^{-1}$.

| | EC measurements + gap-filling + flux partitioning | | | Model simulations | | |
|---|---|---|---|---|---|---|
| Year | NEE | GPP | $R_{ECO}$ | [1]$GPP_{TREES}$ | [1]$R_{TREES\_AG}$ | $R_{TREES\_AG}$ +[2]$R_{FF}$ |
| 2005 | −991 ± 37 | −3816 | 2821 | −2311 | 1033 | 3345 |
| 2006 | −516 ± 114 | −3231 | 2725 | −2590 | 1160 | 3468 |
| 2007 | −952 ± 35 | −4149 | 3207 | −2530 | 1010 | 3122 |
| 2008 | −970 ± 35 | −4023 | 3089 | −2463 | 1007 | 3063 |
| Mean | −857 | −3805 | 2961 | −2473 | 1053 | 3250 |
| as C | −234 | −1038 | 807 | −674 | 287 | 886 |

[1] SPP-model (Mäkelä et al. 2006).

[2] Eq. 1 (App. 3, treatment D)

Table 3. Modelled annual forest floor $CO_2$ effluxes (mean $\pm$ S.E.M.; g $CO_2$ m$^{-2}$ a$^{-1}$) in the four treatments at Kalevansuo peatland. A = peat, B = peat+litter, C = peat+litter+roots and D = peat+litter+roots+ground vegetation. S.E.M is the standard error between the four plot means.

| Year | A | B | C | D |
|------|------|------|------|------|
| 2005 | 1233 $\pm$ 48 | 1745 $\pm$ 121 | 2118 $\pm$ 67 | 2312 $\pm$ 170 |
| 2006 | 1233 $\pm$ 48 | 1741 $\pm$ 108 | 2117 $\pm$ 76 | 2308 $\pm$ 179 |
| 2007 | 822 $\pm$ 40 | n.d. | n.d. | 2112 $\pm$ 141 |
| 2008 | 835 $\pm$ 49 | n.d. | n.d. | 2056 $\pm$ 143 |

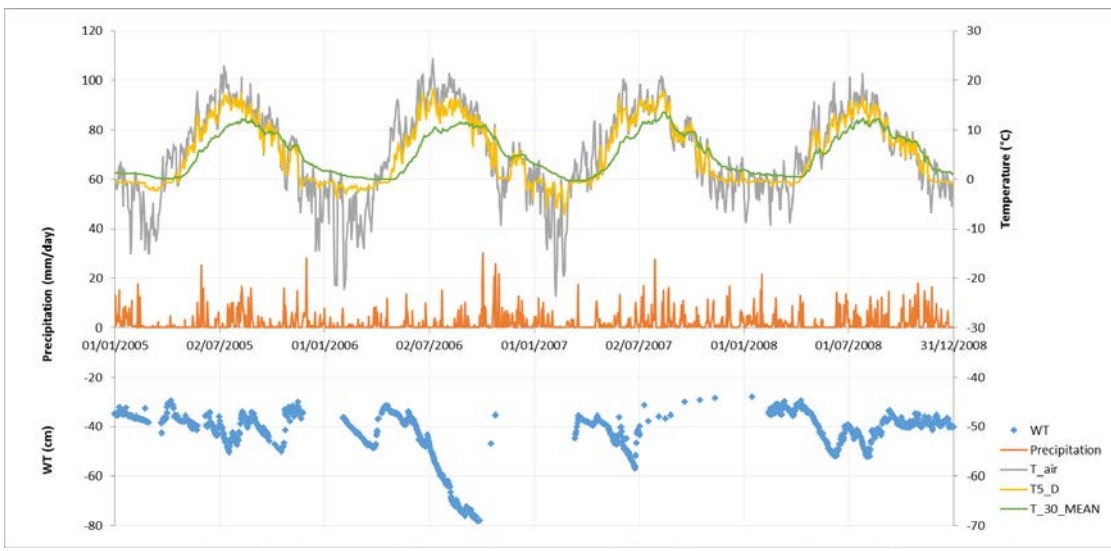

**Figure 1. Daily weather data: Precipitation (average from nearby weather stations), air and soil (5cm, 30 cm) temperatures and average water table level at Kalevansuo peatland.**

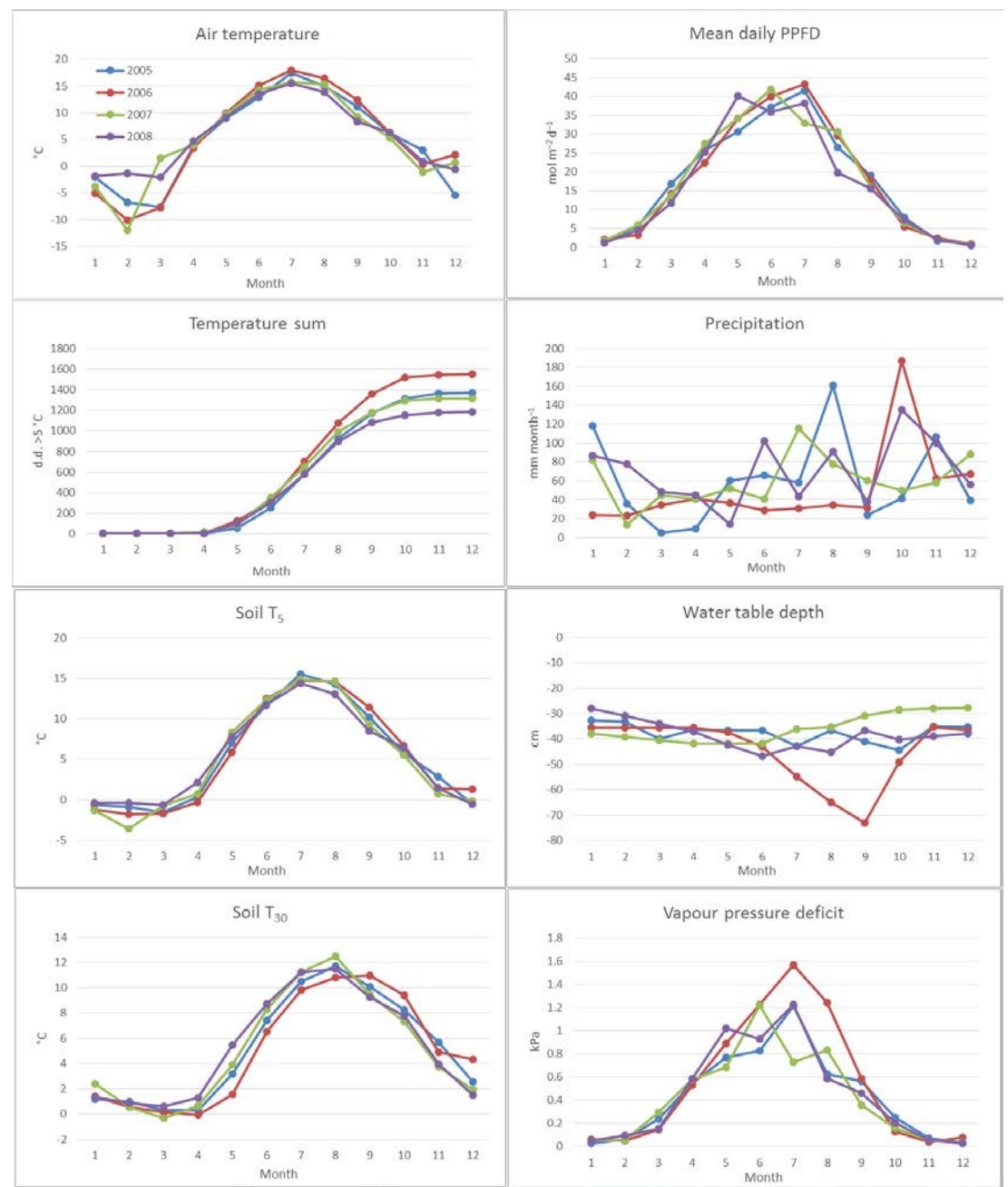

**Figure 2. Weather variables by year and month, measured at Kalevansuo, except precipitation, which is an average from nearby weather stations. Mean daily air temperature (°C) and temperature sum (> 5 °C d.d.) at 2 m height, soil temperatures (°C) at 5 and 30 cm depths, mean daily PPFD (mol m⁻² d⁻¹) at 21.5 m. height, monthly precipitation sum (mm month⁻¹), water table depth (cm) and vapour pressure deficit at 21.5 m. height in the afternoon, 12:00–16:00 local time (kPa).**

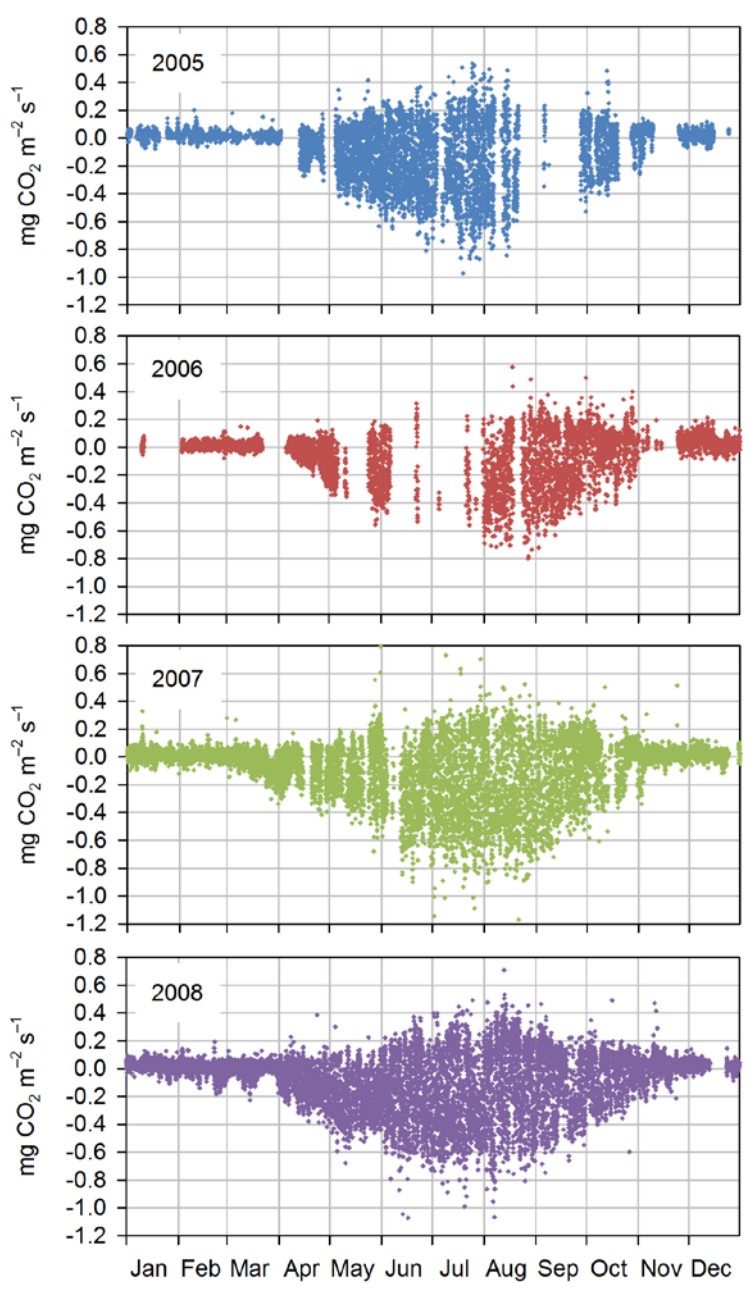

**Figure 3. Quality-controlled half-hourly NEE measured with eddy covariance method at Kalevansuo peatland in 2005–2008.**

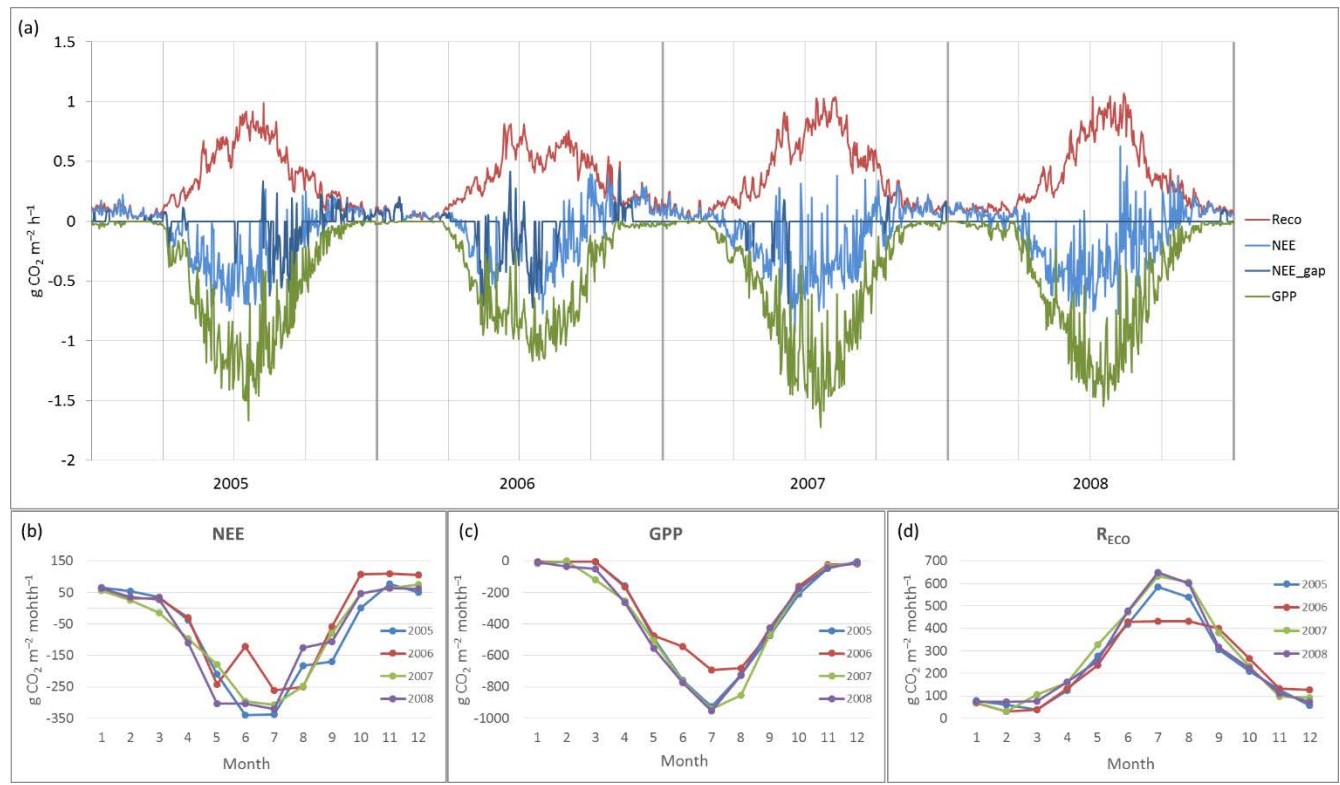

**Figure 4. a) Gap-filled and partitioned daily NEE, GPP and R$_{ECO}$ at Kalevansuo 2005–2008. Full days with missing data shown with dark blue (NEE_gap). b) monthly NEE; c) monthly GPP; and d) monthly R$_{ECO}$.**

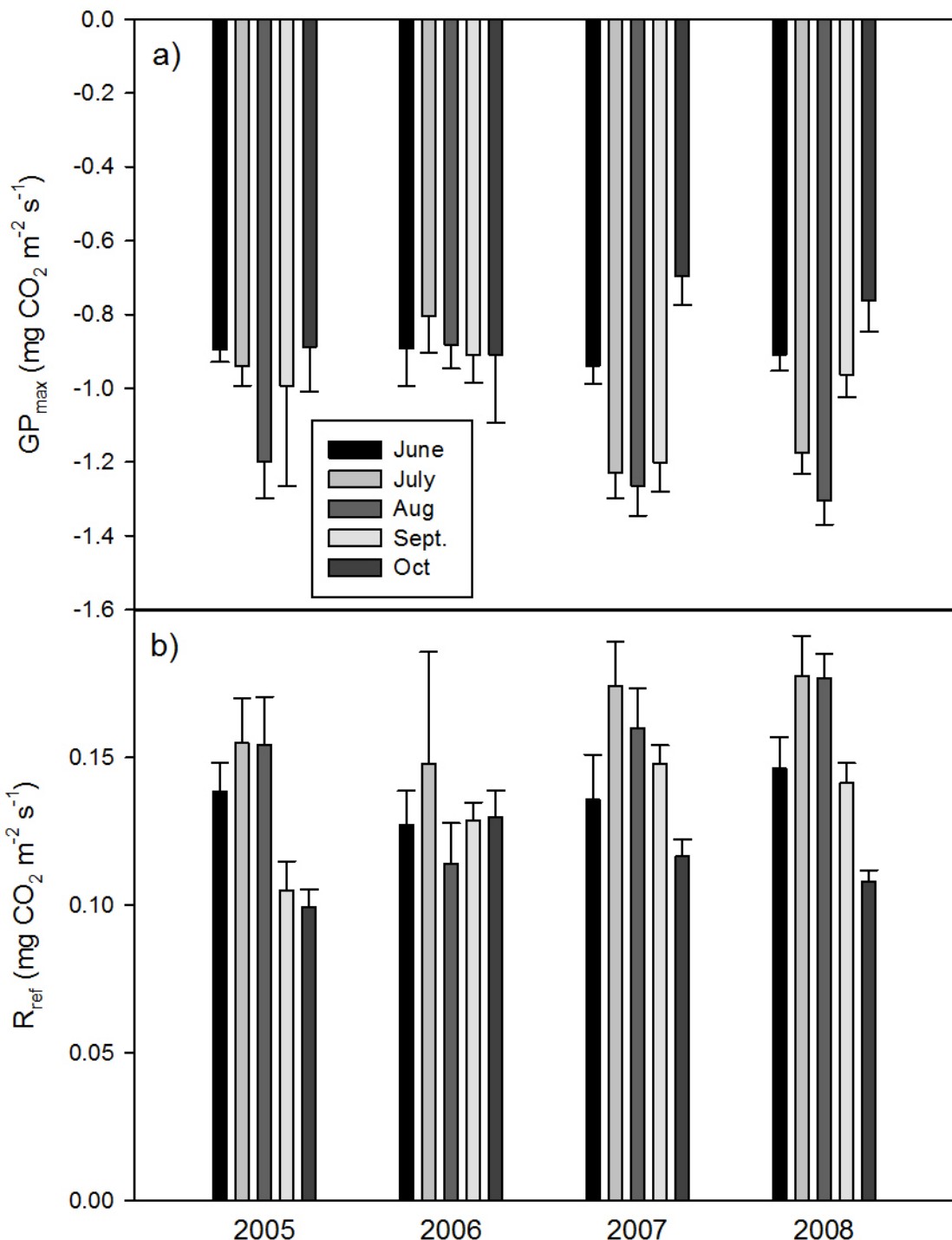

**Figure 5. Parameter values a) $GP_{MAX\,x}$ and b) $R_{REF}$ ± 95% confidence intervals (see Eqs. A2 and A3 in Appendix A, respectively) for June–October in 2005–2008. For the respiration model, a constant value of $E_0$=200K was used, and the GPP model was used here without the VPD term.**

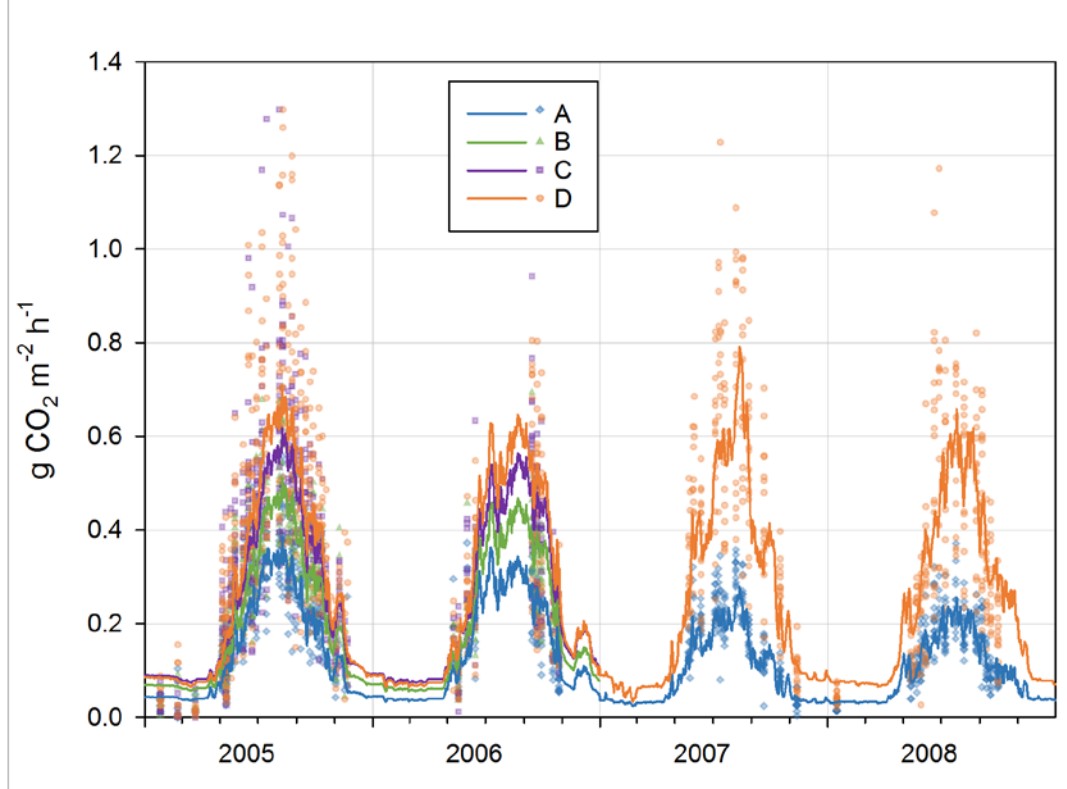

**Figure 6. Forest floor CO₂ efflux from different treatment collars in 2005–2008. A: peat, B: peat+litter, C: peat+litter+roots and D: peat+litter+roots+ground vegetation. Points mark individual chamber measurements and lines modelled daily average fluxes (eq. 1, App. 3). Notice: two individual fluxes (9.8.2005) with values 1.80 (D) and 1.84 (C) are outside the graph range, and were excluded as outliers from the regression models (Table 3).**

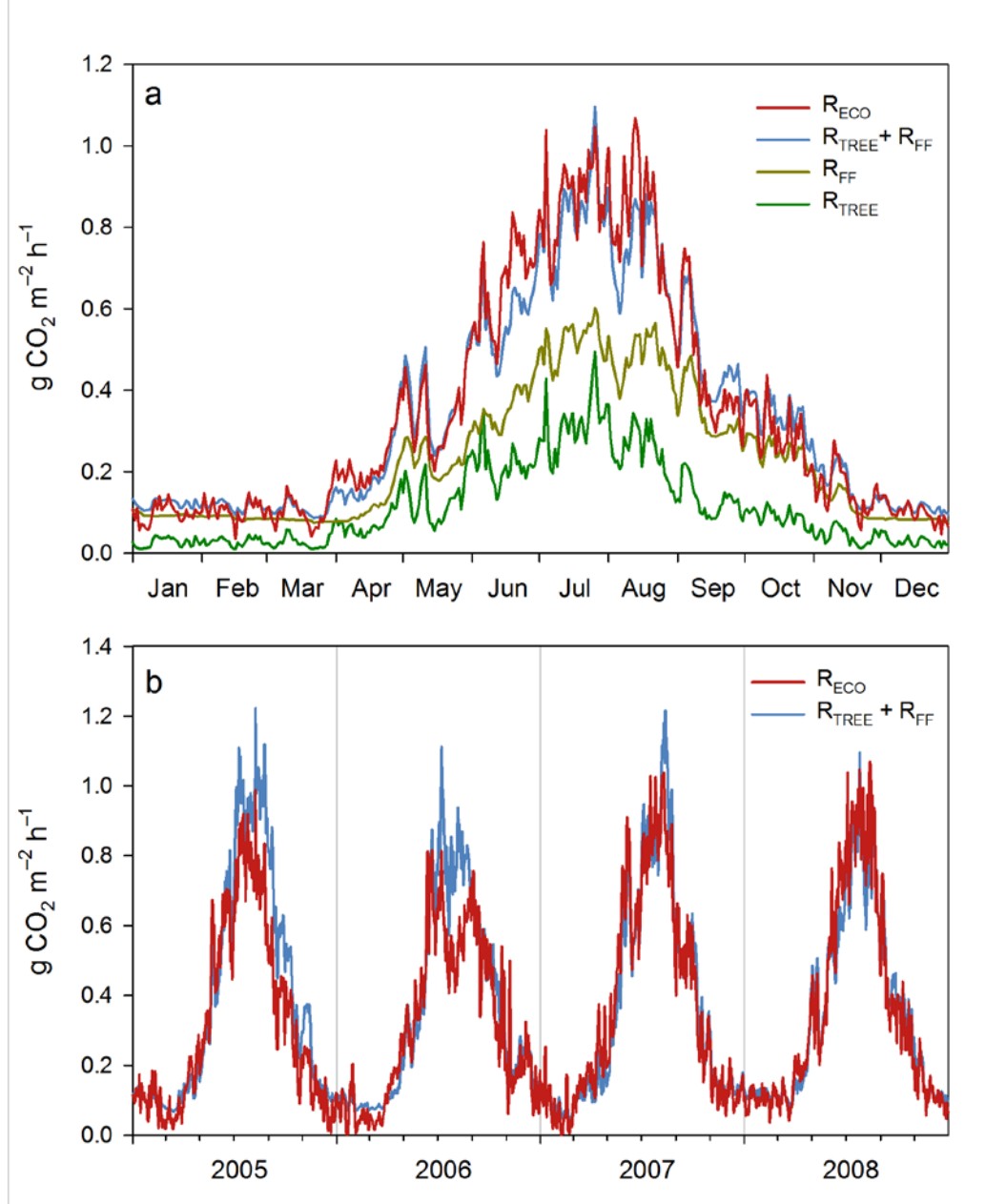

Figure 7. (a) Ecosystem respiration based on measured and gapfilled EC data ($R_{ECO}$) and simulated respiration effluxes of different components at Kalevansuo in 2008. Since $R_{ECO}$ is the total ecosystem respiration, it should equal the sum of above ground respiration of trees ($R_{TREE}$) and total forest floor respiration ($R_{FF}$). $R_{TREE}$ was simulated using the SPP model, while $R_{FF}$ is based on measured flux data and statistical models for the same site and year. (b) $R_{ECO}$ compared with the sum of $R_{TREE}$ + $R_{FF}$ during the whole measurement period 2005–2008.

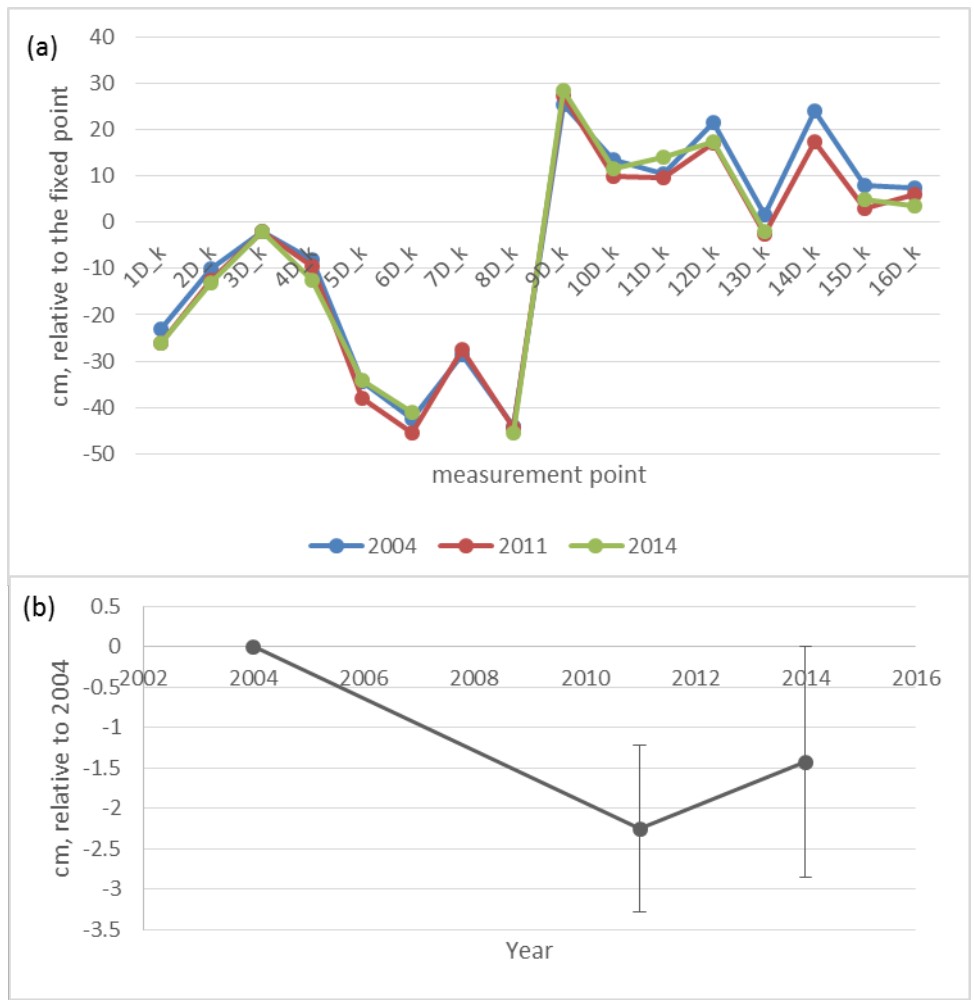

**Figure 8. (a) Elevation of soil surface in 2004, 2011 and 2014 in the middle of the undisturbed D-plots, relative to the fixed benchmark beside the EC-mast. (b) Change in elevation relative to 2004, mean and 2*standard error of the mean. Only the points measured at every occasion are included in the mean and S.E.M values.**

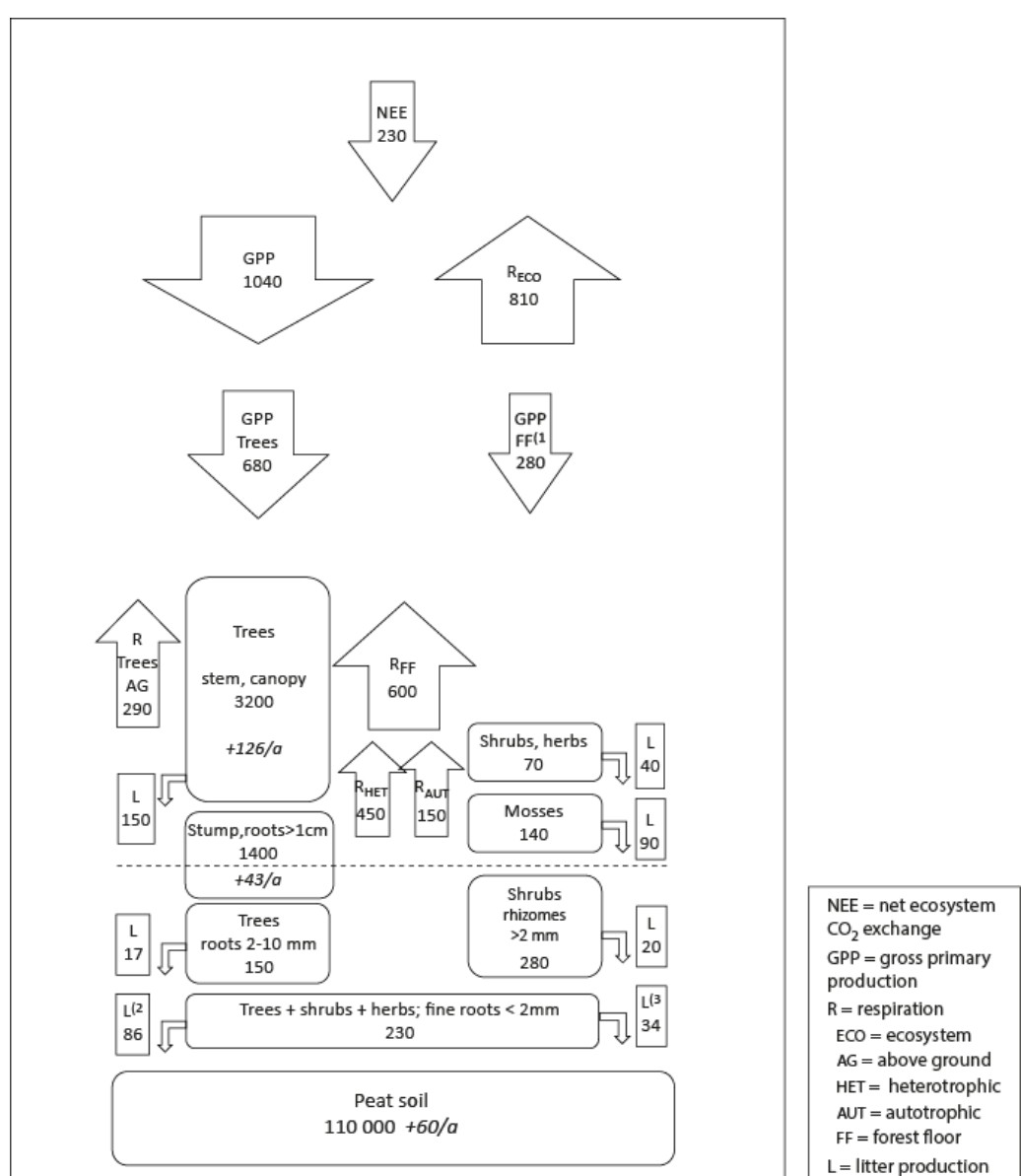

**Figure 9.** Measured carbon pools (rounded boxes; g C m$^{-2}$, and the changes in pools in italics; *g C m$^{-2}$ a$^{-1}$*) and fluxes (arrows and square boxes; g C m$^{-2}$ a$^{-1}$) in Kalevansuo drained peatland. Soil C accumulation is calculated as NEE (230 g C m$^{-2}$ a$^{-1}$) – C sequestration in tree stand biomass (170 g C m$^{-2}$ a$^{-1}$; above- 126 g C m$^{-2}$ a$^{-1}$ and belowground 43 g C m$^{-2}$ a$^{-1}$). Other fluxes and pools are based on measured and modelled values derived from EC and chamber measurements, biomass and litterfall measurements in Kalevansuo. The tree stand biomass is from the fall 2008 measurement.[1] Badorek et al. 2011; [2] Fine root production of trees (Bhuiyan et al. 2016); [3] Fine root production of shrubs and herbs (Bhuiyan et al. 2016).

**Appendix A. Gap-filling and partitioning of net ecosystem exchange**

The gap-filling of the net ecosystem exchange ($NEE$) data obtained from the eddy covariance measurements was performed with the procedures incorporated into the FluxPartFill.py program developed at the Finnish Meteorological Institute. The gap-filling algorithm is based on empirical functions for total ecosystem respiration ($R_{ECO}$) and gross primary production ($GPP$) and thus additionally provides the partitioning of NEE into the RECO and GPP components

$$NEE = R_{ECO} + GPP \tag{A1}$$

Ecosystem respiration was assumed to respond to temperature according to the Arrhenius-type relationship suggested by Lloyd and Taylor (1994)

$$R_{ECO}(T) = R_{REF} \exp\left[E_0\left(\frac{1}{T_{REF}-T_0} - \frac{1}{T-T_0}\right)\right] \tag{A2}$$

where $T$ is temperature, $R_{REF}$ is the reference respiration ($R$ at $T_{REF}$ = 283.15 K), $E_0$ describes the temperature sensitivity of $R$ to $T$, and T0 = 227.13 K is a constant. Eq. (A2) is fitted to nocturnal (photosynthetic photon flux density PPFD < 5 µmol m$^{-2}$ s$^{-1}$) flux data by optimizing the parameters $R_{REF}$ and $E_0$.

Gross primary production was assumed to depend on $PPFD$ according to a rectangular hyperbola that is multiplied by a function $f_{VPD}$ representing the reduction of $GPP$ with increasing water vapour pressure deficit (VPD):

$$GPP(PPFD, VPD) = \frac{\alpha\, PPFD\, GP_{max}}{\alpha\, PPFD + GP_{max}} f_{VPD}(VPD) \tag{A3}$$

where $\alpha$ is the apparent quantum yield and $GP_{MAX}$ is the maximum asymptotic GPP when $f = 1$ ($GPP \to GP_{max}$, as $PPFD \to \infty$). For $f_{VPD}$, we adopted a form that results in $f_{VPD} = 1$ for $VPD \le VPD_1$, $f_{VPD} = f_0$ for $VPD \ge VPD_0$ and a linear reduction from $f_{VPD} = 1$ to $f_0$ between $VPD_1$ and $VPD_0$:

$$f_{VPD}(VPD) = \max\left(f_0, \min\left(1 - \frac{VPD-VPD_1}{VPD_0-VPD_1}(1-f_0)\right)\right) \tag{A4}$$

Eq. (A3) is fitted to daytime ($PPFD$ > 20 µmol m$^{-2}$ s$^{-1}$) net flux data from which the respiration flux, calculated using Eq. (A2) with the optimized parameters, has been subtracted. The Levenberg–Marquardt algorithm as implemented in the LMFIT package (Newville et al. 2014) is used for both $R_{eco}$ and $GPP$ fits.

In FluxPartFill.py, the model parameters are calculated for each day with a centred multiday data window. The length of this window can be made variable (within a specified range) by defining the minimum number of flux and meteorological data that must be available for the fit. In this study, the parameters were fitted using fixed 21- and 11-day windows for $R_{ECO}$ and GPP, respectively. To avoid unrealistic fluctuations in the parameter values due to the multidimensionality of the fitting problem, FluxPartFill.py makes it possible to apply an iterative process in which a varying subset of parameters is fitted in subsequent runs, with an option for manually adjusted parameter time series.

In the present study, $E_0$ was set constant at 200 K for the whole period, based on an initial fit to the whole 4-a data. Air temperature measured at 2 m was used for $T$. For $GPP$, we fitted $\alpha$ and $GP_{MAX}$, while $f_{VPD}$ was based on fixed parameter values: $VPD_1 = 10$ hPa and $VPD_0 = 25$ hPa (Lohila et al. 2011), and $f_0 = 0.4$. Based on the gap-filled time series, using linearly interpolated parameter values where necessary, daily balances are calculated for $NEE$, $R_{ECO}$ and $GPP$.

Newville M., Stensitzki T., Allen D. B. & Ingargiola A. (2014). LMFIT: Non-Linear Least-Square Minimization and Curve-Fitting for Python. doi.org/10.5281/zenodo.11813

## Appendix B. Uncertainty analysis of NEE

The uncertainty of the annual $CO_2$ balance was estimated separately for each year. We followed here the approaches presented by Aurela et al. (2002), Lohila et al. (2011) and Räsänen et al. (2017). The random error arising from the stochastic variability of turbulent fluxes ($E_{MEAS}$) was estimated, similarly to Räsänen et al. (2017), from the difference between the measurements and the corresponding values obtained from the gap-filling model fits (Eqs. A1-A4 in Appendix A). This error varied between 5.8 and 13 g $CO_2$ m$^{-2}$ a$^{-1}$ in 2005-2008 (Table A1). The same approach was applied to the random error arising from the gap-filling of the data ($E_{GAPS}$), which ranged from 6.0 to 7.3 g $CO_2$ m$^{-2}$ a$^{-1}$ in 2005-2008 (Table A1). The uncertainty associated with the corrections for the high-frequency flux loss ($E_{HFL}$) was estimated at 3% of the annual balance (Lohila et al. 2011). For the uncertainty due to the gap-filling of the longer (>2 days) gaps in the flux data ($E_{LONGGAPS}$), we adopted a new approach: as year 2008 had only few gaps, we simulated the impact of longer gaps in other years by assuming similar data gaps in the time series of 2008 and then ran the gap-filling procedures for these compromised data. For each gap, a cumulative $CO_2$ balance was estimated from two differently gap-filled data sets, i.e. the original and the simulated, and the difference of these was assumed to represent the error. The annual error was calculated by assuming that the errors obtained this way for separate gaps were independent of each other. For 2006, a similar simulation was also done with the data of 2005 and 2007, and the average of the three annual estimates obtained was taken as the total error related to the gap-filling of long gaps. In 2008, there was only one longer (10 day) gap in December. The uncertainty due to this gap was calculated by adopting the daily errors estimated for December 2007, resulting in an $E_{LONGGAPS}$ of 13.3 g $CO_2$ m$^{-2}$ a$^{-1}$ for 2008. The total uncertainty in the annual balance was calculated by assuming that $E_{MEAS}$, $E_{GAPS}$ and $E_{LONGGAPS}$ are independent.

Table B1. Uncertainty analysis of the annual $CO_2$ balance, NEE (g $CO_2$ m$^{-2}$ a$^{-1}$). Error components are explained in the text, and $n_{obs}$ denotes the number of accepted flux observations in 2005-2008.

|  | 2005 | 2006 | 2007 | 2008 |
| --- | --- | --- | --- | --- |
| $E_{MEAS}$ | 5.8 | 6.3 | 12.9 | 13.0 |
| $E_{GAPS}$ | 6.2 | 6.5 | 7.3 | 6.0 |
| $E_{HFL}$ | 29.7 | 15.5 | 28.6 | 29.1 |
| $E_{LONGGAPS}$ | 20.8 | 113 | 13.2 | 13.3 |
| $n_{obs}$ | 7868 (44.9%) | 7240 (41.3%) | 11678 (66.7%) | 12842 (73.3%) |
| NEE ± error | −991 ± 37 | −516 ± 114 | −952 ± 35 | −970 ± 35 |

The total error of ±37 g $CO_2$ m$^{-2}$ a$^{-1}$ in 2005 was significantly reduced from that (± 100 g m$^{-2}$ a$^{-1}$) reported by Lohila et al. (2011). This was mainly due to the different approach adopted for the error estimates for the compensation of long data gaps: for this, Lohila et al. (2011) shifted model parameters 2 weeks forward and backward, which resulted in a relative error of 10.7% of the annual balance. We consider the present approach more realistic, as it is based on assessing the effect of realized gaps on actual measurement data. However, it is obvious that the uncertainty estimate for 2006 is limited by the fact that the summer of that year was exceptionally dry, and the changes in NEE induced by the drought cannot be accurately estimated based on the data of 2008. It is likely that the dynamics of photosynthesis and respiration during a dry summer are different from a normal year. This hypothesis gains support from the observation that the $R_{ECO}$ and GPP parameters in August 2006 differed markedly from those estimated for the other years (Fig. 5).

As a further sensitivity test, we estimated the most conservative range for the NEE uncertainty in summer 2006. Because our gap-filling model fills the long gaps by linear interpolation, the outcome depends on the selection of the start and end points of the interpolation. The sensitivity test was performed by assuming different parameter scenarios during the longest gap in the parameters (13 June – 4 August, during which there were only 212 valid NEE observations available). In one scenario, a parameter had the starting point value during the whole gap, while in the other it dropped immediately to the level at the end point and stayed there over the whole gap. The annual NEE was calculated for each combination of different $R_{REF}$ and $GP_{MAX}$ dynamics. The two most extreme cases (either both $R_{REF}$ and $GP_{MAX}$ were reduced for the whole gap, or both were increased) produced an annual NEE of -377 and -844 g $CO_2$ m$^{-2}$ s$^{-1}$, respectively. These can be considered as the most conservative estimates of annual NEE, and thus we can safely conclude that the annual NEE was negative in 2006, i.e. the ecosystem acted as a $CO_2$ sink even during the exceptionally dry year.

Despite the large gaps during the growing season of 2006, and the large uncertainty resulting from these, the annual NEE balance of 2006 differed significantly from the other years. This difference between two annual balances ($NEE_i$ and $NEE_{i+1}$) was considered significant if the 95% confidence interval of the difference, defined as

$$(NEE_{i+1} - NEE_i) \pm 2\sqrt{SE_{i+1}^2 + SE_i^2}$$ (Eq. B1)

where $SE_i$ is the standard error of $NEE_i$, did not cross zero.

**Appendix C.** Parameter values of forest floor $CO_2$ efflux models (Eq.1) for different collar treatments (A = peat, B = peat + litter, C = peat + litter + roots and D = peat + litter + roots + ground vegetation) and years. $R_{REF5}$ and $R_{REF30}$ are respirations at 10 °C for the 5 cm and 30 depths (g $CO_2$ m$^{-2}$ h$^{-1}$), $E_{05}$ and $E_{030}$ are is temperature sensitivities of respiration for the same layers, $r^2$ is coefficient of determination of the model and n is the number of observations.

| Treatment | Plot | Years | $R_{REF5}$ | $E_{05}$ | $R_{REF30}$ | $E_{030}$ | $r^2$ | n |
|---|---|---|---|---|---|---|---|---|
| A | 1 | 2005–2006 | 0.151 | 302.4 | 0.088 | 716.4 | 0.76 | 127 |
| A | 1 | 2007–2008 | 0.124 | 319.5 | 0.027 | 755.2 | 0.78 | 108 |
| A | 2 | 2005–2006 | 0.138 | 359.7 | 0.072 | 754.5 | 0.77 | 105 |
| A | 2 | 2007–2008 | 0.104 | 431.5 | 0.021 | 418.2 | 0.90 | 108 |
| A | 3 | 2005–2006 | 0.127 | 324.6 | 0.087 | 1011.4 | 0.77 | 113 |
| A | 3 | 2007–2008 | 0.120 | 299.6 | 0.015 | 508.6 | 0.79 | 107 |
| A | 4 | 2005–2006 | 0.086 | 432.6 | 0.123 | 297.7 | 0.62 | 114 |
| A | 4 | 2007–2008 | 0.112 | 331.1 | 0.027 | 511.7 | 0.64 | 102 |
| B | 1 | 2005–2006 | 0.175 | 284.7 | 0.142 | 660.2 | 0.75 | 135 |
| B | 2 | 2005–2006 | 0.159 | 282.6 | 0.199 | 560.7 | 0.72 | 121 |
| B | 3 | 2005–2006 | 0.183 | 244.6 | 0.146 | 1053.0 | 0.67 | 125 |
| B | 4 | 2005–2006 | 0.102 | 303.7 | 0.183 | 584.6 | 0.82 | 127 |
| C | 1 | 2005–2006 | 0.237 | 338.0 | 0.163 | 942.3 | 0.77 | 145 |
| C | 2 | 2005–2006 | 0.152 | 193.3 | 0.217 | 532.9 | 0.49 | 131 |
| C | 3 | 2005–2006 | 0.139 | 311.9 | 0.228 | 605.9 | 0.58 | 124 |
| C | 4 | 2005–2006 | 0.204 | 240.8 | 0.261 | 824.9 | 0.46 | 137 |
| D | 1 | 2005–2006 | 0.165 | 339.3 | 0.202 | 741.3 | 0.50 | 130 |
| D | 1 | 2007–2008 | 0.171 | 360.5 | 0.146 | 885.8 | 0.54 | 107 |
| D | 2 | 2005–2006 | 0.192 | 361.4 | 0.295 | 593.3 | 0.70 | 129 |
| D | 2 | 2007–2008 | 0.195 | 383.2 | 0.254 | 518.9 | 0.71 | 105 |
| D | 3 | 2005–2006 | 0.050 | 581.4 | 0.319 | 411.4 | 0.56 | 129 |
| D | 3 | 2007–2008 | 0.123 | 261.0 | 0.300 | 586.8 | 0.58 | 106 |
| D | 4 | 2005–2006 | 0.265 | 280.4 | 0.280 | 1115.6 | 0.62 | 121 |
| D | 4 | 2007–2008 | 0.229 | 367.7 | 0.146 | 550.1 | 0.56 | 103 |