# Peer review of "Persistent carbon sink at a boreal drained bog forest"

_Biogeosciences, 2017_

## Referee Comment (RC1) · Anonymous Referee #1 · 30 Jan 2018

General comment: The authors present a convincing dataset and sound reasoning in a comprehensive study which will be without doubt controversially discussed. The results are convincing, but I still find it puzzling to read about C accumulation rates of this magnitude without seeing a moss or shrub layer developing. Therefore, I think that the authors should spend more effort in discussing possible remains of the stored soil carbon. Is there possibly a reallocation of rhizosphere or subsoil organic carbon? Is the bulk density increasing? Are soil pores being filled with particulate organic matter? It the assumption that export of DOC/POC is negligible, reasonable? Specific comments: I appreciate that, for site and methods information, the reader is referred to Lohila et al. (2011). However, even if this means repeating things, I suggest adding ba-

sic information on a couple of things: When and how was the stand set up? How deep and how strongly decomposed is the peat? Please give some basic information on peat properties. This is important for the discussion that, I think should be somewhat enhanced. Were chamber measurements done with opaque chambers to prevent photosynthesis by shrubs? How high were the chambers to make shrubs fit inside them? 2.2.3 How can you be sure that 7 CH4 flux calculations in a 6 month period are sufficient to derive a correct estimate for CH4 efflux? How can you assume that CH4 fluxes reported by Lohila (2011) can be applied to a later time? I think that it is not correct to apply measurements done at an earlier time for other time periods, unless properly modelled. This is not decisive for the main message of the manuscript but should still be described some more. P 7, L 16: A "high NEE" implies a high CO2 uptake, but this is not the case here. Please avoid this expression here. You could say "lower net CO2 uptake", if this is what you mean to say. P 14, L9: "making up", not "making" P11, L24-25: I am not so sure that, when trees die, root carbon becomes part of the soil C pool. After harvest, there is usually large C release from this C pool. P12, L7: That depends whether you assume root carbon to remain in the soil following harvesting. Last paragraph on page 12: Exactly, this remains to be seen! P13, L3-4: I don't think that your data justifies saying that the drained peatland stores AT LEAST as much C as undisturbed peatlands. Are you referring to the mean long term C storage of all peatlands here? You are able to conclude that, in a very limited time of the lifecycle of the rotation, there is a soil C sink, even under meteorologically variable seasons. Figure 2: Please don't display precipitation by drawing straight lines between the months. A bar graph is much more adequate.

---

## Referee Comment (RC2) · Anonymous Referee #2 · 4 Feb 2018

GENERAL COMMENTS This is a well written, clear paper that tackles the important yet controversial topic of peat accumulation of peatlands drained for silviculture. The results of the experiments presented in the manuscript are interesting in that they highlight that the ecosystem is still a sink, in spite of having been drained. The authors approach the problem with rigour, and measure gas exchange via two means, eddy covariance and chambers. They report the results of their experiments in a clear manner and discuss possible implications. I especially thought the last section in the discussion "Can the carbon sink last?" was valuable. In this section they discuss the ultimate fate of the carbon and how important the carbon accumulation in the soil (rather than in the forest biomass) is for sustainable practices. The authors present an important

topic at a time where peatland ecosystem services are being valued more than ever before and management practices need to be evidence-driven.

SPECIFIC COMMENTS My main criticism of the manuscript is the issue of DOC losses and actual long-term carbon accumulation (as measured in the peat profile using dating and carbon measurements).

1. Is DOC loss really a minor component – due to ineffective ditching and high transpiration? All peatlands lose some carbon via DOC, the losses in pristine peatlands are often non-negligible. Drained peatlands may experience even larger (or sometimes smaller) DOC losses. Are the authors suggesting there is no water that comes off their site due to evapotranspiration? Can the authors show evidence of studies on forested drained peatlands that have measured DOC losses during high precipitation events after a dry period, for example? Evidence from non-forested drained peatlands suggests losses are substantial (see for example papers from Strack et al., 2008 and others).

1. Page 10 Line 24: Again, this issue is related to DOC loss. A large Rhet is originated in the top part of peatlands when these are pristine (as was the case for the study site in the reference the authors cite, Chimner and Cooper, 2003). Once a peatland has been drained for a long time, I am not sure this necessarily holds true for every peatland. Of course, labile carbon from the top of the peat is preferentially decomposed always, but the older peat might also get decomposed if exposed for a peatlalong time and this may be lost via DOC preferentially (not completely oxidised as CO2 and therefore not detected by any of the means the authors deployed). See for example Evans, C. D., et al. (2014), Contrasting vulnerability of drained tropical and high-latitude nds to fluvial loss of stored carbon, Global Biogeochem. Cycles, 28, doi:10.1002/2013GB004782.

2. An important methodological question is, considering the uncertainty in the measurements (NEE has an error between 35-114 g CO2 m-2 yr-1) and the fact that the authors did not measure DOC losses, I wonder if coring the peat, and measuring recent accumulation of peat directly by dating the peat would have been a good way

of validating the idea of the carbon sink. It is a shame that this has not been done, since the authors even cored the peat at the site. Have any of those cores been analysed for carbon and dated? Is peat actually accumulating (and shrinking) at the same time? The authors present a very interesting result, but with the uncertainty in NEE and the possibility of DOC losses, one cannot be absolutely certain that this site is actually accumulating peat in the long term. Additionally, the authors need to be careful of estimating actual peat accumulation via modern fluxes alone. Experiments in which both a) fluxes and b) peat C accumulation measured via dating of the peat profile have been carried out, have found a large disparity between the two measurements. In fact, experiments demonstrate long term peat accumulation is much smaller than contemporary fluxes suggest, see: "Contemporary carbon fluxes do not reflect the long-term carbon balance for an Atlantic blanket bog", JoshuaÂăRatcliffe,ÂăRoxaneÂăAndersen,ÂăRussellÂăAnderson,ÂăAnthonyÂăNewton,ÂăDavidÂăCampbell,ÂăDmitriÂăMauquoy,ÂăR HoloceneÂăVol 28, Issue 1, pp. 140 – 149, First PublishedÂăJune 30, 2017, https://doi.org/10.1177/0959683617715689 I'd be really interested in seeing some of those collected cores dated from Kalevansuo.

I am not expecting the authors to expand their methodology to measure DOC or C accumulation in the peat profile, but I perhaps expect them to discuss these two main issues a little more in their manuscript. This is important since the title suggests that C accumulation is the main topic of the article, but actually peat C accumulation has not been directly measured.

Additionally. I have some suggestions for clarifying the methodology used: - How long before the measurements were the chambers installed? I suggest this is included in the methods. - Why are the chambers 30 cm deep? Did the authors know roots were not significant lower down? Other studies have used 50 or 60 cm deep chambers. - Repeated clipping – did you clip before every measurement? Or if not, then how often? Please consider including this in the methods. - Was the biomass of the understorey measured? I thought it was according to the methods, but then in Page 11 line 11
the authors write that they assume biomass to not be increasing. Was the biomass only measured once? If so, this needs to be made explicit in the methods section. "Thus the method should not be overestimating soil C pool increase, more likely 15 underestimating it. " It is unclear to the reader why this should be so, could the authors explain more explicitly in the text? - Why are the errors estimated (a very nice treatment of uncertainty in Appendix 2) but not reported in the main text? (What figure is (Fig. xx in chapter 3.2) in Appendix 2?)

TECHNICAL CORRECTIONS Page 1: Line 10: turn into (instead of turn to) Line 11: We measured the carbon ("the" is missing) Line 11: NEE of CO2 was measured with an eddy covariance method... Line 14: Biomass (in singular, not plural) and litter production.... Line 14-16: Consider changing to: Soil balance was estimated...from NEE, and this showed that the soil itself was a carbon sink as well. Page 6: Line 34: Consider changing to: WT varied also spatially ("quite much" should be deleted)..... Page 10: Line 33: Consider changing to: "The value is considerably smaller than that reported for agricultural fields."

---

## Referee Comment (RC3) · Anonymous Referee #3 · 6 Feb 2018

On my first read of this paper I thought it was an excellent study. I still do but with one major caveat, which I outline below in detail.

The authors are correct that the commonly used EF by the IPCC for drainage of organic soils do not apply to the case of forested peatlands, particularly the drained forests of Finland. The forest management practices there are well done and the drainage is not excessive. Also the ditches are very clean so that also minimizes emissions. The same authors have published aspects of this story before. Their publications and this study support an argument for Tier 2 and 3 methods applied to the Finnish drained forests.

The present manuscript develops the argument further that drained mires for forest

remain sinks even in a year of extreme drought. This would be an important proof of the continued sink potential. Unfortunately, on my second read I realized that the only year where there are no eddy covariance measurements during the growing season was the year of the significant drought - 2096. The authors show the large, persistent water table drop – it is far in excess of any other year and the duration of the drought in exceptional unique in their data set. To accept the evidence that is used to support the authors' main conclusion one has to believe that the gap-filled techniques are appropriate for drought conditions. I doubt this very much. The authors evaluate their model for 'normal' years but they provide no evidence that the model applies for the drought conditions experienced in 2006. The actual test is very weak – they examine annual NEP and NECB. It is simulated reasonably well but this is largely irrelevant as the key test is to see if the model simulates GPP and Reco under drought. The reader has to know how well the model performs for extremes, particularly how well its captures the influence of high VPDs and lower water tables. Most gap-filling algorithms do well for average conditions – this is what they are designed to do. However, like most ecosystem models they do very poorly when the conditions deviate beyond the normal range of variance. The water table during 2006 was much deeper than anything experienced during the other three years. Further, the drop in water table is a persistent secular trend. The authors provide no evidence that their gap-filling algorithms can simulate GPP or Reco under these conditions. The model has no dependency on water availability. The GPP model has VPD and PPFD so the reduction in water availability has to come through the VPD function but there is no link. However, it appears that the VPDs experienced in the drought were unique – i.e. well outside the range that was used to develop the parameters in the model. I assume GPmax is derived for normal conditions not drought conditions.

For me to believe their results I would need to see the following:

1, A demonstration that the parameters in their LUE model ($\alpha$, GPmax) apply to drought conditions; 2. How Reco is sensitive to changes in water storage, or how temperature

reflects the effect of low water contents; 3. the range of conditions that the data was used to fit the gap-filling functions and how for outside that ranges were the conditions in 2006; and 4. an error analysis that looks at the performance of the model functions with extreme conditions – firstly using measurement and simulated data within the range of conditions observed, and hten by sensitivity analysis. The sensitivity analysis will not have any moisture effects in it so maybe an examination of the model the authors use versus what other models that incorporate moisture produce.

With these four steps the authors could place the 2006 data on solid ground or determine that their main conclusions are an artifact of the gap-filling approach. Without this the paper is simply infer untested functions for conditions they were never designed to handle and the key argument of the paper collapses.

Given the that central argument of the manuscript is dependent on the 2006 Fluxes and these are not measurements but gap-filled the authors have to provide proof the gap-filling is appropriate. If the authors can provide this proof then this manuscript would be useful addition to the literature and it would have policy relevance. Without the proof of the gap-filling validity the evidence supporting the main conclusion collapses.

———————————————————

---

## Author Comment (AC1) · 14 Mar 2018

"General comment: The authors present a convincing dataset and sound reasoning in a comprehensive study which will be without doubt controversially discussed. The results are convincing, but I still find it puzzling to read about C accumulation rates of this magnitude without seeing a moss or shrub layer developing. Therefore, I think that the authors should spend more effort in discussing possible remains of the stored soil carbon."

[Figure]

–Moss and shrubs are still present in the peatland (p. 2, r 35-38) and they are growing vigorously, producing a lot of litter (p. 11, r. 17-22).

"Is there possibly a reallocation of rhizosphere or subsoil organic carbon?"

–Yes, very probably. Root growth is a very important input of C into the peat soil (p. 11, r. 20).

–Add: When decomposed, part of the released C is translocated as a solute into deeper peat layers (Domisch et al. 2000).

"Is the bulk density increasing? "

–We do not have bulk density measurements from Kalevansuo peatland prior to drainage, but compared to similar pristine sites (38 kg m-3 natural treed fens (Minkkinen and Laine 1998), bulk density of the surface 0-20 cm layer is higher in Kalevansuo (94 kg m-3 ). It is therefore likely that bulk density has incresed after drainage, as "is evident in all drained peatlands (e.g. Minkkinen and Laine 1998)" (p. 11, r. 2).

"Are soil pores being filled with particulate organic matter?"

–We have no observations of this phenomenon. However, this is one of the consequences of decomposition, leading towards higher bulk density.

"Is the assumption that export of DOC/POC is negligible, reasonable?"

–Leaching of DOC, i.e. the output of dissolved C from Finnish drained peatlands varies between 10 to 15 g C m−2 a−1 (Sallantaus and Kaipainen, 1996; Kortelainen et al., 1997; Sarkkola et al., 2009; Rantakari et al., 2010). This is 4–7% of the estimated NEE and 17–25% of soil C balance in Kalevansuo. The smaller the soil C balance the higher the share of DOC export naturally becomes. However, the input of DOC into forest soils is of the same magnitude as the output. According to Lindroos et al. (2008) Finnish forest soils receive 2–6 g C m−2 a−1 dissolved C as deposition in stand throughfall and in 2–10 g C m−2 a−1 percolation water. Kalevansuo is mainly

ombrotrophic, so that DOC in percolation water is insignificant. However, as mentioned earlier, ditches in Kalevansuo are ineffective and transpiration of the tree stand is likely an important pathway for water output in such a case like Kalevansuo (Sarkkola et al. 2010) decreasing the possibility for big DOC losses. It is thus likely that net export of DOC does not have a major significance for the C balance at Kalevansuo peatland.

"I appreciate that, for site and methods information, the reader is referred to Lohila et al. (2011). However, even if this means repeating things, I suggest adding basic information on a couple of things: When and how was the stand set up? How deep and how strongly decomposed is the peat? Please give some basic information on peat properties. This is important for the discussion that, I think should be somewhat enhanced."

–It is not entirely clear if the referee means the history of the tree stand, or the setup of measurements. The description of both can of course be added, e.g.:

–The site has been naturally forested long before drainage as evidenced by seemingly very old scattered stumps of Scots pine found in all parts of the site. Tree ages of the present Scots pine stand, as determined in 2005 from increment cores extending to tree pith of some sample trees at the height of 1.3m and assuming 15 years to reach that height, varied from 67 to 179 years with an average of 120 years (n=7).

–The site was set up for C flux measurements in June-August 2004. Collars were inserted, litter collected and plants removed from the treated collar plots in June 2004. The eddy covariance tower was erected in August 2004.

–Peat depth, measured from the 33 sample plots varied from 1.3 to 3.0 metres, average being 2.2 m. Mean peat bulk density was 94 kg m-3 in the 0-20 cm layer.

–The peat (prior to drainage) in the area of EC footprint is composed mainly of the remains of Sphagna (Sphagnum fuscum, S. magellanicum), Ericaceous shrubs and cottongrass (Eriophorum vaginatum) (Mathijssen et al. 2017). After drainage the remains of forest mosses and woody roots have increased their share in surface peat. Drainage has increased surface peat oxidation which is seen as a shallow layer of more decomposed peat about 10-20 cm below surface. Remains of several forest fires are also present especially in the surface layers 30-50 cm, where a charcoal layer is clearly visible, but also in the deeper layers from 70 to 180 cm. Forest fires have thus significantly decreased the long-term rate of C accumulation into the Kalevansuo peatland (Mathijssen et al. 2017).

"Were chamber measurements done with opaque chambers to prevent photosynthesis by shrubs? How high were the chambers to make shrubs fit inside them?"

–Chamber measurements were done with opaque chambers to measure dark respiration. In this study photosynthesis was not measured. $CO_2$ chamber was 14.9 cm high. In plots with vascular plants we used 5 cm and 10 cm extra collars to fit the plants in the chamber. The chamber volume was corrected accordingly.

"2.2.3 How can you be sure that 7 $CH_4$ flux calculations in a 6 month period are sufficient to derive a correct estimate for $CH_4$ efflux? How can you assume that $CH_4$ fluxes reported by Lohila (2011) can be applied to a later time? I think that it is not correct to apply measurements done at an earlier time for other time periods, unless properly modelled. This is not decisive for the main message of the manuscript but should still be described some more."

–Methane flux on drained peatlands is quite well studied. Fluxes from strips are always small, often negative (see e.g. the synthesis of wetland $CH_4$ emissions by Turetsky et al. 2014). Kalevansuo is not an exception in this rule. Annual variation was small, average flux estimates from the highest flux months July-September were almost same in the years 2004 (-11.04 mg m-2 month-1) and 2005 (-11.98 mg m-2 month-1). It is improbable that other years would have remarkably changed the flux. The new measurements from ditches were from a short period, but they did cover the high and low flux seasons (Minkkinen and Laine 2006) and they revealed that accounting ditches

did not significantly change the areal flux estimate. The total CH4 flux remained negative. Much higher emissions from ditches would have been needed to change the conclusions. As carbon, the CH4 fluxes are negligible compared to all other C fluxes in Kalevansuo. Although modelling CH4 fluxes would be interesting, it has no relevance for this study.

"P 7, L 16: A "high NEE" implies a high CO2 uptake, but this is not the case here. Please avoid this expression here. You could say "lower net CO2 uptake", if this is what you mean to say."

–We have tried to use NEE unambiguously so that negative NEE indicates C uptake and positive net emission (p. 4, r.10). Thus high NEE means high CO2 emission. We can however change this expression to "lower net CO2 uptake" as suggested.

"P 14, L9: "making up", not "making" "

–OK. Changed accordingly.

"P11, L24-25: I am not so sure that, when trees die, root carbon becomes part of the soil C pool. After harvest, there is usually large C release from this C pool. "

–This is conceptual debate. We mean that when trees are cut, roots and the C in their structures, remain in the soil until they are decomposed. Thus, they temporarily become part of the soil C pool. Here we refer to roots > 1cm which would not be as fast decomposed as the fine roots.

"P12, L7: That ["soil is currently a C sink"] depends whether you assume root carbon to remain in the soil following harvesting. Last paragraph on page 12: Exactly, this remains to be seen! "

–The word "currently" refers to the current situation, when trees continue growing and cuttings have not been done. Situation after a possible harvesting is another issue, as discussed later. (p. 12).

"P13, L3-4: I don't think that your data justifies saying that the drained peatland stores AT LEAST as much C as undisturbed peatlands. Are you referring to the mean long term C storage of all peatlands here? You are able to conclude that, in a very limited time of the lifecycle of the rotation, there is a soil C sink, even under meteorologically variable seasons. "

–OK. Yes this was based on long-term accumulation values from natural peatlands, which is (according to Clymo's (1984) conceptual model of peat bog growth, and supported by many observations (e.g. Tolonen and Turunen 1996, Clymo et al. 1998, Turunen et al. 2002)) in bogs bigger than the actual, current accumulation. However, as the actual accumulation rates are not accurately known for either natural mires or for Kalevansuo peatland, we will change "at least as much" to "similar to".

"Figure 2: Please don't display precipitation by drawing straight lines between the months. A bar graph is much more adequate. "

–We agree that bar graph is usually more adequate for such a discontinuous data like monthly precipitation. However, in this case with the same legend in multiple graphs, we think that line drawing is a better choice. It is easy to quickly see the differences between years in all variables.

References

Clymo, R.S. 1984. The limits to peat bog growth. Phil. Trans. R. Soc. Lond. Biol. Sci. 303: 605-654.

Clymo, R.S., Turunen, J., Tolonen, K. 1998. Carbon accumulation in peatland, Oikos, 81, 368-388.

Kortelainen, P., Saukkonen, S., and Mattsson, T.: Leaching of nitrogen from forested catchments in Finland, Global Biogeochem. Cy., 11, 627–638, 1997.

Lindroos, A.-J., Derome, J., Mustajärvi, K., NÂÍojd, P., Beuker, E., and Helmisaari, H.-S.: Fluxes of dissolved organic carbon in stand throughfall and percolation water in 12

boreal coniferous stands on mineral soils in Finland, Boreal Environ. Res., 13B, 22–34, 2008.

Lohila, A, Minkkinen, K., Aurela, M., Tuovinen, J-P., Penttilä, T., and Laurila, T. 2011. Greenhouse gas flux measurements in a forestry-drained peatland indicate a large carbon sink. Biogeosciences 8: 3203–3218. http://dx.doi.org/ 10.5194/bg-8-3203-2011

Mathijssen, P. J. H., Kähkölä, N., Tuovinen, J.-P., Lohila, A., Minkkinen, K., Laurila T. and Väliranta, M. 2017. Lateral expansion and carbon exchange of a boreal peatland in Finland resulting in 7000 years of positive radiative forcing, Journal of Geophysical Research: Biogeosciences 122. http://dx.doi.org/10.1002/2016JG003749.

Minkkinen, K. & Laine, J. 2006. Vegetation heterogeneity and ditches create spatial variability in methane fluxes from peatlands drained for forestry. Plant and Soil: 289–304.

Rantakari, M., Mattsson, T., Kortelainen, P., Piirainen, S., Fin′er, L., and Ahtiainen, M.: Organic and inorganic carbon concentrations and fluxes from managed and unmanaged boreal first-order catchments, Sci. Total Environ., 408, 1649–1658, 2010.

Sallantaus, T. and Kaipainen, H.: Water-carried element balances of peatlands, in: Northern peatlands in global climatic change, edited by: Laiho, R., Laine, J., and Vasander, H., Publications of the Academy of Finland, Edita, Helsinki, 197–203, 1996.

Sarkkola et al. 2010. Role of tree stand evapotranspiration in maintaining satisfactory drainage conditions in drained peatlandsCan. J. For. Res. 40: 1485–1496 (2010) doi:10.1139/X10-084

Sarkkola, S., Koivusalo, H., Laur′en, A., Kortelainen, P., Mattsson, T., Palviainen, M., Piirainen, S., Starr, M., and Fin′er, L.: Trends in hydrometeorological conditions and stream water organic carbon in boreal forested catchments, Sci. Total Environ., 408, 92–101, 2009.

Turetsky, et al. 2014. A synthesis of methane emissions from 71 northern,

temperate, and 1 subtropical wetlands. Global Change Biology 20: 2183–2197. http://dx.doi.org/10.1111/gcb.12580

Turunen J., Tomppo E., Tolonen K. & Reinikainen A. 2002. Es timating carbon accumulation rates of undrained mires in Finland – application to boreal and subarctic regions. Holocene 12(1): 69–80.

---

## Author Comment (AC2) · 14 Mar 2018

"GENERAL COMMENTS This is a well written, clear paper that tackles the important yet controversial topic of peat accumulation of peatlands drained for silviculture. The results of the experiments presented in the manuscript are interesting in that they highlight that the ecosystem is still a sink, in spite of having been drained. The authors approach the problem with rigour, and measure gas exchange via two means, eddy covariance and chambers. They report the results of their experiments in a clear manner and discuss possible implications. I especially thought the last section in the discussion "Can the carbon sink last?" was valuable. In this section they discuss the ultimate fate of the carbon and how important the carbon accumulation in the soil (rather than in the forest biomass) is for sustainable practices. The authors present an important topic at a time where peatland ecosystem services are being valued more than ever before and management practices need to be evidence-driven. "

"SPECIFIC COMMENTS My main criticism of the manuscript is the issue of DOC losses and actual long-term carbon accumulation (as measured in the peat profile using dating and carbon measurements). "

"Is DOC loss really a minor component – due to ineffective ditching and high transpiration? All peatlands lose some carbon via DOC, the losses in pristine peatlands are often non-negligible. Drained peatlands may experience even larger (or sometimes smaller) DOC losses. Are the authors suggesting there is no water that comes off their site due to evapotranspiration? Can the authors show evidence of studies on forested drained peatlands that have measured DOC losses during high precipitation events after a dry period, for example? Evidence from non-forested drained peatlands suggests losses are substantial (see for example papers from Strack et al., 2008 and others). "

–The same answer as for referee #1: Leaching of DOC, i.e. the output of dissolved C from Finnish drained peatlands varies between 10 to 15 g C m−2 a−1 (Sallantaus and Kaipainen, 1996; Kortelainen et al., 1997; Sarkkola et al., 2009; Rantakari et al., 2010). This is 4–7% of the estimated NEE and 17–25% of soil C balance in Kalevansuo. The smaller the soil C balance the higher the share of DOC export naturally becomes. However, the input of DOC into the forest soils is of the same magnitude as the output. According to Lindroos et al. (2008) Finnish forest soils receive 2–6 g C m−2 a−1 dissolved C as deposition in stand throughfall and in 2–10 g C m−2 a−1 percolation water. Kalevansuo is mainly ombrotrophic, so that DOC in percolation water is insignificant. However, as mentioned earlier, ditches in Kalevansuo are ineffective and transpiration of the tree stand is likely an important pathway for water output in such a case like Kalevansuo (Sarkkola et al. 2010) decreasing the possibility for big DOC

losses. It is thus likely that net export of DOC does not have a major significance for the C balance at Kalevansuo peatland.

"Page 10 Line 24: Again, this issue is related to DOC loss. A large Rhet is originated in the top part of peatlands when these are pristine (as was the case for the study site in the reference the authors cite, Chimner and Cooper, 2003). Once a peatland has been drained for a long time, I am not sure this necessarily holds true for every peatland. Of course, labile carbon from the top of the peat is preferentially decomposed always, but the older peat might also get decomposed if exposed for a peat along time and this may be lost via DOC preferentially (not completely oxidised as CO2 and therefore not detected by any of the means the authors deployed). See for example Evans, C. D., et al. (2014), Contrasting vulnerability of drained tropical and high-latitude nds to fluvial loss of stored carbon, Global Biogeochem. Cycles, 28, doi:10.1002/2013GB004782. "

–We understand this possibility. We added discussion about DOC (see above).

"2. An important methodological question is, considering the uncertainty in the measurements (NEE has an error between 35-114 g CO2 m-2 yr-1) and the fact that the authors did not measure DOC losses, I wonder if coring the peat, and measuring recent accumulation of peat directly by dating the peat would have been a good way of validating the idea of the carbon sink. It is a shame that this has not been done, since the authors even cored the peat at the site. Have any of those cores been analysed for carbon and dated? Is peat actually accumulating (and shrinking) at the same time? The authors present a very interesting result, but with the uncertainty in NEE and the possibility of DOC losses, one cannot be absolutely certain that this site is actually accumulating peat in the long term. Additionally, the authors need to be careful of estimating actual peat accumulation via modern fluxes alone. Experiments in which both a) fluxes and b) peat C accumulation measured via dating of the peat profile have been carried out, have found a large disparity between the two measurements. In fact, experiments demonstrate long term peat accumulation is much smaller than contemporary fluxes suggest, see: "Contemporary carbon fluxes do not reflect the long-term carbon balance

for an Atlantic blanket bog", Joshua Ratcliffe, Roxane Andersen, Russell Anderson, Anthony Newton, David Campbell, Dmitri Mauquoy,Holocene Vol 28, Issue 1, pp. 140 – 149, First Published June 30, 2017, https://doi.org/10.1177/0959683617715689 I'd be really interested in seeing some of those collected cores dated from Kalevansuo. "

–Kalevansuo has been cored extensively and historical C accumulation has been determined using radiocarbon dating (Mathijssen et al. 2017). However, drainage is such a recent happening (35 yrs before our study) that 14C dating is not a reliable method to determine post-drainage C accumulation in peat. Even if the surface peat could be dated accurately, root growth into deeper layers would mess up the results. Many different peat-coring-dating-pollen-bulk density-based methods have been tried (e.g. Kruger et al. 2016, Minkkinen and Laine 1998, Minkkinen et al. 1999, Simola et al. 2012, Turetsky et al. 2004) but they have large uncertainties. Eddy covariance combined with biomass growth measurements is the most accurate method for this purpose.

–We know that long-term peat accumulation rates may be very different from current ones, although sometimes they may be quite similar, if the measurement span is long enough (e.g. Aurela et al. 2004, Roulet et al. 2007, Nilsson et al. 2008). It is also well-known that long-term average rates, determined by peat coring and radiocarbon dating, are not the same as the actual, current (or decadal average) rates. Modelling is needed to estimate that (e.g. Clymo et al. 1998, Frolking et al. 2014). However, as we have discussed, we are not trying to estimate long-term peat accumulation, only the current one.

"I am not expecting the authors to expand their methodology to measure DOC or C accumulation in the peat profile, but I perhaps expect them to discuss these two main issues a little more in their manuscript. This is important since the title suggests that C accumulation is the main topic of the article, but actually peat C accumulation has not been directly measured. "

"Additionally. I have some suggestions for clarifying the methodology used: How long before the measurements were the chambers installed? I suggest this is included in the methods. "

–Collars were installed in June 2004, half a year before the respiration data was used here.

"Why are the chambers 30 cm deep? Did the authors know roots were not significant lower down? Other studies have used 50 or 60 cm deep chambers. "

–In this site average WT is so high that we estimated that 30 cm deep collars are deep enough. It is however possible that roots have grown under the collars, and thus Rpeat would be overestimated.

"Repeated clipping – did you clip before every measurement? Or if not, then how often? Please consider including this in the methods. "

–Plants were clipped every time before measurements if new plants had emerged.

"Was the biomass of the understorey measured? I thought it was according to the methods, but then in Page 11 line 11 the authors write that they assume biomass to not be increasing. Was the biomass only measured once? If so, this needs to be made explicit in the methods section. "

–Biomass of ground vegetation was measured once (p. 5, r. 3) only.

"11,14 "Thus the method should not be overestimating soil C pool increase, more likely underestimating it. " It is unclear to the reader why this should be so, could the authors explain more explicitly in the text? "

–The correlation between tree stand and ground vegetation biomass is negative (Reinikainen et al. 1984): as the tree stand grows bigger, the available resources (light, nutrients) for ground vegetation decrease, leading, in theory, to decreased growth and biomass. If this is correct for Kalevansuo, C pool in ground vegetation would have decreased, since tree stand C has increased. As dCsoil = NEE - dC biomass (all marked positive here), decrease in biomass C would lead to increase in soil C. However, we now think that this sentence is overinterpretation, and will delete the "more likely underestimating it".

"Why are the errors estimated (a very nice treatment of uncertainty in Appendix 2) but not reported in the main text? (What figure is (Fig. xx in chapter 3.2) in Appendix 2?)"

–Fig. xx on page 33, line 31 should refer to Fig. 5

–We will add the following text about the errors to the results section:

–p.7, line 11: "…the annual NEE was surprisingly similar in other years, varying from –950 to –990 g CO2 m–2 a–1. The estimated uncertainty in the annual budget, including the random error in measurements and gap-filling, the uncertainty in the correction for high-frequency loss and the uncertainty related to gap-filling of the longer than 2-day gaps, varied from 35 to 114 g m–2 a–1, corresponding to 3.6 to 22% of the respective annual balance.

–Fig. 5c, d.

"TECHNICAL CORRECTIONS Page 1: Line 10: turn into (instead of turn to) "

–corrected

"Line 11: We measured the carbon ("the" is missing) "

–added

"Line 11: NEE of CO2 was measured with an eddy covariance method. . . "

–"the" added

"Line 14: Biomass (in singular, not plural) and litter production. . .. "

–ok

"Line 14-16: Consider changing to: Soil balance was estimated. . .from NEE, and this showed that the soil itself was a carbon sink as well. "

–ok, changed

"Page 6: Line 34: Consider changing to: WT varied also spatially ("quite much" should be deleted). "

–ok, deleted

". ... Page 10: Line 33: Consider changing to: "The value is considerably smaller than that reported for agricultural fields." "

–ok, changed

References:

Aurela et al. 2004. The timing of snow melt controls the annual CO2 balance in a subarctic fen Geophysical Research Letters, Vol. 31, L16119, doi:10.1029/2004GL020315, 2004

Clymo, R.S., Turunen, J., Tolonen, K. 1998. Carbon accumulation in peatland, Oikos, 81, 368-388.

Frolking et al. 2014. Exploring the relationship between peatland net carbon balance and apparent carbon accumulation rate at century to millennial time scales. The Holocene 2014, Vol. 24(9) 1167–1173. DOI: 10.1177/0959683614538078

Kortelainen, P., Saukkonen, S., and Mattsson, T.: Leaching of nitrogen from forested catchments in Finland, Global Biogeochem. Cy., 11, 627–638, 1997.

Krüger, J.P., Alewell, C., Minkkinen, K., Szidat, S. and Leifeld, J. 2016. Calculating carbon changes in peat soils drained for forestry with four different profile-based methods. Forest Ecology and Management 381: 29–36. http://dx.doi.org/10.1016/j.foreco.2016.09.006

Lindroos, A.-J., Derome, J., MustajÂÍarvi, K., NÂÍojd, P., Beuker, E., and Helmisaari, H.-S.: Fluxes of dissolved organic carbon in stand throughfall and percolation water in 12 boreal coniferous stands on mineral soils in Finland, Boreal Environ. Res., 13B, 22–34, 2008.

Mathijssen, P. J. H., Kähkölä, N., Tuovinen, J.-P., Lohila, A., Minkkinen, K., Laurila T. and Väliranta, M. 2017. Lateral expansion and carbon exchange of a boreal peatland in Finland resulting in 7000 years of positive radiative forcing, Journal of Geophysical Research: Biogeosciences 122. http://dx.doi.org/10.1002/2016JG003749.

Minkkinen, K. and Laine, J. 1998. Long term effect of forest drainage on the peat carbon stores of pine mires in Finland. Can. J. For. Res: 28: 1267–1275.

Minkkinen, K., Vasander, H., Jauhiainen, S., Karsisto, M. and Laine, J. 1999. Post-drainage changes in vegetation composition and carbon balance in Lakkasuo mire, Central Finland. Plant and Soil 207:107–120.

Nilsson et al. 2008. Contemporary carbon accumulation in a boreal oligotrophic minerogenic mire – a significant sink after accounting for all C-fluxes. Global Change Biology (2008) 14, 1–16, doi: 10.1111/j.1365-2486.2008.01654.x

Rantakari, M., Mattsson, T., Kortelainen, P., Piirainen, S., Fin′er, L., and Ahtiainen, M.: Organic and inorganic carbon concentrations and fluxes from managed and unmanaged boreal first-order catchments, Sci. Total Environ., 408, 1649–1658, 2010.

Reinikainen, A., Vasander, H. and Lindholm, T. 1984. Plant biomass and primary production of southern boreal mire-ecosystems in Finland. In: Proceedings of the 7th International Peat Congress, Dublin,Ireland, 18-23 June, 1984, vol. 4. The Irish National Peat Committee, Dublin. pp. 1-20

Roulet et al. 2007. Contemporary carbon balance and late Holocene carbon accumulation in a northern peatlandGlobal Change Biology (2007) 13, 397–411, doi: 10.1111/j.1365-2486.2006.01292.x

Sallantaus, T. and Kaipainen, H.: Water-carried element balances of peatlands, in: Northern peatlands in global climatic change, edited by: Laiho, R., Laine, J., and Vasander, H., Publications of the Academy of Finland, Edita, Helsinki, 197–203, 1996.

Sarkkola et al. 2010. Role of tree stand evapotranspiration in maintaining satisfactory drainage conditions in drained peatlandsCan. J. For. Res. 40: 1485–1496 (2010) doi:10.1139/X10-084

Sarkkola, S., Koivusalo, H., Laur′en, A., Kortelainen, P., Mattsson, T., Palviainen, M., Piirainen, S., Starr, M., and Fin′er, L.: Trends in hydrometeorological conditions and stream water organic carbon in boreal forested catchments, Sci. Total Environ., 408, 92–101, 2009.

Simola, H., Pitkänen, A. & Turunen, J. (2012) Carbon loss in drained forestry peatlands in Finland, estimated by re-sampling peatlands surveyed in the 1980s. European Journal of Soil Science 63: 798–807.

Turetsky et al. 2004. Dating recent peat deposits. Wetlands, 24(2): 324–356.

---

## Author Response (AR1)

*"General comment: The authors present a convincing dataset and sound reasoning in a comprehensive study which will be without doubt controversially discussed. The results are convincing, but I still find it puzzling to read about C accumulation rates of this magnitude **without seeing a moss or shrub layer developing**. Therefore, I think that the **authors should spend more effort in discussing possible remains of the stored soil carbon.**"*

–Moss and shrubs are still present in the peatland (p. 2, r 35-38) and they are growing vigorously, producing a lot of litter (p. 11, r. 17-22).

"Is there possibly a reallocation of rhizosphere or subsoil organic carbon?"

–Yes, very probably. Root growth is a very important input of C into the peat soil (p. 11, r. 20).

–Add: When decomposed, part of the released C is translocated as a solute into deeper peat layers (Domisch et al. 2000).

"Is the bulk density increasing? "

– We do not have bulk density measurements from Kalevansuo peatland prior to drainage, but compared to similar pristine sites (38 kg m$^{-3}$ natural pine mires (Minkkinen and Laine 1998), bulk density of the surface 0-20 cm layer is higher in Kalevansuo (94 kg m$^{-3}$ ). It is therefore likely that bulk density has increased in Kalevansuo after drainage." (p. 11, r. 2).

"Are soil pores being filled with particulate organic matter?"

–We have no observations of this phenomenon. However, this is one of the consequences of decomposition, leading towards higher bulk density.

"Is the assumption that export of DOC/POC is negligible, reasonable?"

– Taking into account the leaching of C would have only a minor effect on the NEE estimate. We do not have dissolved organic carbon (DOC) measurements from Kalevansuo, but leaching of DOC, i.e. the output of dissolved C, from Finnish drained peatlands ranges from 10 to 15 g C m−2 a−1 (Sallantaus and Kaipainen, 1996; Kortelainen et al., 1997; Sarkkola et al., 2009; Rantakari et al., 2010). This is 4–7% of the estimated NEE and 17–25% of the soil C balance at Kalevansuo. As the ditches in Kalevansuo are ineffective and the transpiration of the tree stand and ground vegetation is an important pathway for water output (Sarkkola et al. 2010), leaching of DOC at Kalevansuo is likely at the lower end of the observed range. Thus, taking leaching into account would not change the conclusion on soil C sink.

"I appreciate that, for site and methods information, the reader is referred to Lohila et al. (2011). However, even if this means repeating things, I suggest adding basic information on a couple of things: When and how was the stand set up? How deep and how strongly decomposed is the peat? Please give some basic information on peat properties. This is important for the discussion that, I think should be somewhat enhanced."

–It is not entirely clear if the referee means the history of the tree stand, or the setup of measurements. The description of both can of course be added, e.g.:

– The site has been naturally forested long before drainage as evidenced by very old scattered stumps of Scots pine found in all parts of the site. Tree ages of the present Scots pine stand, as determined in 2005 from increment cores of sample trees (n=7), varied from 67 to 179 years with an average of 120 years..

–The site was set up for C flux measurements in June-August 2004. Collars were inserted, litter collected and plants removed from the treated collar plots in June 2004. The eddy covariance tower was erected in August 2004.

– Peat depth, measured from the 33 sample plots varies from 1.3 to 3.0 metres, average being 2.2 m. Mean peat bulk density is 94 kg m$^{-3}$ in the 0–20 cm layer. The peat accumulated prior to drainage at the study area is composed mainly of the remains of Sphagna (*Sphagnum fuscum, S. magellanicum*), Ericaceous shrubs and cottongrass (*Eriophorum vaginatum*) (Mathijssen et al. 2017). After drainage the remains of forest mosses and woody roots have increased their share in surface peat. Drainage has increased surface peat oxidation which is seen as a shallow layer of more decomposed peat about 10–20 cm below surface. Remains of several forest fires are also present especially in the surface layers 30–50 cm, where a charcoal layer is clearly visible, but also in the deeper layers from 70 to 180 cm.

"Were chamber measurements done with opaque chambers to prevent photosynthesis by shrubs? How high were the chambers to make shrubs fit inside them?"

–Chamber measurements were done with opaque chambers to measure dark respiration. In this study photosynthesis was not measured. $CO_2$ chamber was 14.9 cm high. In plots with vascular plants we used 5 cm and 10 cm extra collars to fit the plants in the chamber. The chamber volume was corrected accordingly.

"2.2.3 How can you be sure that 7 CH4 flux calculations in a 6 month period are sufficient to derive a correct estimate for CH4 efflux? How can you assume that CH4 fluxes reported by Lohila (2011) can be applied to a later time? I think that it is not correct to apply measurements done at an earlier time for other time periods, unless properly modelled. This is not decisive for the main message of the manuscript but should still be described some more."

–Methane flux on drained peatlands is quite well studied. Fluxes from strips are always small, often negative (see e.g. the synthesis of wetland CH4 emissions by Turetsky et al. 2014). Kalevansuo is not an exception in this rule. Annual variation was small, average flux estimates from the highest flux months July-September were almost same in the years 2004 (-11.04 mg m$^{-2}$ month$^{-1}$) and 2005 (-11.98 mg m$^{-2}$ month$^{-1}$). It is improbable that other years would have remarkably changed the flux. The new measurements from

ditches were from a short period, but they did cover the high and low flux seasons (Minkkinen and Laine 2006) and they revealed that accounting ditches did not significantly change the areal flux estimate. The total CH4 flux remained negative. Much higher emissions from ditches would have been needed to change the conclusions. As carbon, the $CH_4$ fluxes are negligible compared to all other C fluxes in Kalevansuo. Although modelling CH4 fluxes would be interesting, it has no relevance for this study.

"P 7, L 16: A "high NEE" implies a high CO2 uptake, but this is not the case here. Please avoid this expression here. You could say "lower net CO2 uptake", if this is what you mean to say."

–We have tried to use NEE unambiguously so that negative NEE indicates C uptake and positive net emission (p. 4, r.10). Thus high NEE means high CO2 emission. In October-December NEE was always either zero or positive, so we cannot really say "lower net CO2 uptake" here.

"P 14, L9: "making up", not "making" "

–OK. Changed accordingly.

"P11, L24-25: I am not so sure that, when trees die, root carbon becomes part of the soil C pool. After harvest, there is usually large C release from this C pool. "

–This is conceptual debate. We mean that when trees are cut, roots and the C in their structures, remain in the soil until they are decomposed. Thus, they temporarily become part of the soil C pool. Here we refer to roots > 1cm which would not be as fast decomposed as the fine roots.

"P12, L7: That ["soil is currently a C sink"] depends whether you assume root carbon to remain in the soil following harvesting. Last paragraph on page 12: Exactly, this remains to be seen! "

–The word "currently" refers to the current situation, when trees continue growing and cuttings have not been done. Situation after a possible harvesting is another issue, as discussed later. (p. 12).

"P13, L3-4: I don't think that your data justifies saying that the drained peatland stores AT LEAST as much C as undisturbed peatlands. Are you referring to the mean long term C storage of all peatlands here? You are able to conclude that, in a very limited time of the lifecycle of the rotation, there is a soil C sink, even under meteorologically variable seasons. "

–OK. Yes this was based on long-term accumulation values from natural peatlands, which is (according to Clymo's (1984) conceptual model of peat bog growth, and supported by many observations (e.g. Tolonen and Turunen 1996, Clymo et al. 1998, Turunen et al. 2002)) in bogs bigger than the actual, current accumulation. However, as the actual accumulation rates are not accurately known for either natural mires or for Kalevansuo peatland, we will change "at least as much" to "similar to".

"Figure 2: Please don't display precipitation by drawing straight lines between the months. A bar graph is much more adequate. "

–We agree that bar graph is usually more adequate for such a discontinuous data like monthly precipitation. However, in this case with the same legend in multiple graphs, we think that line drawing is a better choice. It is easy to quickly see the differences between years in all variables.

**Anonymous Referee #2

"GENERAL COMMENTS This is a well written, clear paper that tackles the important yet controversial topic of peat accumulation of peatlands drained for silviculture. The results of the experiments presented in the manuscript are interesting in that they highlight that the ecosystem is still a sink, in spite of having been drained. The authors approach the problem with rigour, and measure gas exchange via two means, eddy covariance and chambers. They report the results of their experiments in a clear manner and discuss possible implications. I especially thought the last section in the discussion "Can the carbon sink last?" was valuable. In this section they discuss the ultimate fate of the carbon and how important the carbon accumulation in the soil (rather than in the forest biomass) is for sustainable practices. The authors present an important topic at a time where peatland ecosystem services are being valued more than ever before and management practices need to be evidence-driven. "

"SPECIFIC COMMENTS My main criticism of the manuscript is the issue of DOC losses and actual long-term carbon accumulation (as measured in the peat profile using dating and carbon measurements). "

"Is DOC loss really a minor component – due to ineffective ditching and high transpiration? All peatlands lose some carbon via DOC, the losses in pristine peatlands are often non-negligible. Drained peatlands may experience even larger (or sometimes smaller) DOC losses. Are the authors suggesting there is no water that comes off their site due to evapotranspiration? Can the authors show evidence of studies on forested drained peatlands that have measured DOC losses during high precipitation events after a dry period, for example? Evidence from non-forested drained peatlands suggests losses are substantial (see for example papers from Strack et al., 2008 and others). "

–**The same answer as for referee #1**: Taking into account the leaching of C would have only a minor effect on the NEE estimate. We do not have dissolved organic carbon (DOC) measurements from Kalevansuo, but leaching of DOC, i.e. the output of dissolved C, from Finnish drained peatlands ranges from 10 to 15 g C m$^{-2}$ a$^{-1}$ (Sallantaus and Kaipainen, 1996; Kortelainen et al., 1997; Sarkkola et al., 2009; Rantakari et al., 2010). This is 4–7% of the estimated NEE and 17–25% of the soil C balance at Kalevansuo. As the ditches in Kalevansuo are ineffective and the transpiration of the tree stand and ground vegetation is an important pathway for water output (Sarkkola et al. 2010), leaching of DOC at Kalevansuo is likely at the lower end of the observed range. Thus, taking leaching into account would not change the conclusion on soil C sink.

"Page 10 Line 24: Again, this issue is related to DOC loss. A large Rhet is originated in the top part of peatlands when these are pristine (as was the case for the study site in the reference the authors cite, Chimner and Cooper, 2003). Once a peatland has been drained for a long time, I am not sure this necessarily holds true for every peatland. Of course, labile carbon from the top of the peat is preferentially decomposed always, but the older peat might also get decomposed if exposed for a peat along time and this may be lost via DOC preferentially (not completely oxidised as CO2 and therefore not detected by any of the means the authors deployed). See for example Evans, C. D., et al. (2014), Contrasting vulnerability of drained tropical and high-latitude nds to fluvial loss of stored carbon, Global Biogeochem. Cycles, 28, doi:10.1002/2013GB004782. "

–We understand this possibility. We added discussion about DOC (see above).

"2. An important methodological question is, considering the uncertainty in the measurements (NEE has an error between 35-114 g CO2 m-2 yr-1) and the fact that the authors did not measure DOC losses, I wonder if coring the peat, and measuring recent accumulation of peat directly by dating the peat would have been a good way of validating the idea of the carbon sink. It is a shame that this has not been done, since the authors even cored the peat at the site. Have any of those cores been analysed for carbon and dated? Is peat actually accumulating (and shrinking) at the same time? The authors present a very interesting result, but with the uncertainty in NEE and the possibility of DOC losses, one cannot be absolutely certain that this site is actually accumulating peat in the long term. Additionally, the authors need to be careful of estimating actual peat accumulation via modern fluxes alone. Experiments in which both a) fluxes and b) peat C accumulation measured via dating of the peat profile have been carried out, have found a large disparity between the two measurements. In fact, experiments demonstrate long term peat accumulation is much smaller than contemporary fluxes suggest, see: "Contemporary carbon fluxes do not reflect the long-term carbon balance for an Atlantic blanket bog", Joshua Ratcliffe, Roxane Andersen, Russell Anderson, Anthony Newton, David Campbell, Dmitri Mauquoy,Holocene Vol 28, Issue 1, pp. 140 – 149, First Published June 30, 2017, https://doi.org/10.1177/0959683617715689 I'd be really interested in seeing some of those collected cores dated from Kalevansuo. "

– Based on four-year NEE and tree growth data, we estimated that the accumulation of C in soil was on average 60 g C $m^{-2}$ $a^{-1}$ during the four-year period. Since the tree stand growth data is based on five-year average, we cannot say whether the soil C balance has been positive in all the studied years. Neither can we say if the long-term soil C balance of the peatland would stay similar in the future. In natural mires, where long-term peat C accumulation can be reliably estimated from peat coring and radiocarbon ($^{14}$C) dating, the multi-year mean NEE derived from EC measurements has typically been similar to the long-term accumulation rate (Aurela et al. 2004, Roulet et al. 2007, Nilsson et al. 2008), although not in all cases (Ratcliffe et al. 2017). It is well-known that long-term average rates, determined by peat coring and radiocarbon dating, are not necessarily the same as the actual, current (or decadal average) rates (e.g. Clymo et al. 1998, Frolking et al. 2014).

– Kalevansuo has been cored extensively, and historical C accumulation has been determined using radiocarbon dating (Mathijssen et al. 2017). However, the drainage took place so recently (35 years before our study) that post-drainage C accumulation cannot be reliably determined using $^{14}$C dating. Even if the surface peat could be dated accurately, root growth into deeper layers would mess up the C accumulation estimate. Several peat-coring methods have been tried to estimate post-drainage changes in peat C stocks (e.g. Kruger et al. 2016, Minkkinen and Laine 1998, Minkkinen et al. 1999, Simola et al. 2012, Turetsky et al. 2004) but they all have large uncertainties. However, as discussed above, we were not trying to estimate long-term peat accumulation, only the current rate. Eddy covariance combined with biomass growth measurements is the most accurate method for this purpose.

"I am not expecting the authors to expand their methodology to measure DOC or C accumulation in the peat profile, but I perhaps expect them to discuss these two main issues a little more in their manuscript. This is important since the title suggests that C accumulation is the main topic of the article, but actually peat C accumulation has not been directly measured. "

–Peat C accumulation was not the main topic, but the ecosystem C balance. The title has been changed to better reflect the main topic.

"Additionally. I have some suggestions for clarifying the methodology used:

How long before the measurements were the chambers installed? I suggest this is included in the methods. "

–Collars were installed in June 2004, half a year before the respiration data was used here.

"Why are the chambers 30 cm deep? Did the authors know roots were not significant lower down? Other studies have used 50 or 60 cm deep chambers. "

–In this site average WT is so high that we estimated that 30 cm deep collars are deep enough. It is however possible that roots have grown under the collars, and thus Rpeat would be overestimated.

"Repeated clipping – did you clip before every measurement? Or if not, then how often? Please consider including this in the methods. "

–Plants were clipped every time before measurements if new plants had emerged.

"Was the biomass of the understorey measured? I thought it was according to the methods, but then in Page 11 line 11 the authors write that they assume biomass to not be increasing. Was the biomass only measured once? If so, this needs to be made explicit in the methods section. "

–Biomass of ground vegetation was measured once (p. 5, r. 3) only.

"11,14 "Thus the method should not be overestimating soil C pool increase, more likely underestimating it. " It is unclear to the reader why this should be so, could the authors explain more explicitly in the text? "

–The correlation between tree stand and ground vegetation biomass is negative (Reinikainen et al. 1984): as the tree stand grows bigger, the available resources (light, nutrients) for ground vegetation decrease, leading, in theory, to decreased growth and biomass. If this is correct for Kalevansuo, C pool in ground vegetation would have decreased, since tree stand C has increased. As dCsoil = NEE - dC biomass (all marked positive here), decrease in biomass C would lead to increase in soil C. However, we now think that this sentence is overinterpretation, and deleted the "more likely underestimating it".

"Why are the errors estimated (a very nice treatment of uncertainty in Appendix 2) but not reported in the main text? (What figure is (Fig. xx in chapter 3.2) in Appendix 2?)"

–Fig. xx on page 33, line 31 should refer to Fig. 5

–We added the following text about the errors to the results section:

–p.7, line 21: "With the exception of the dry year of 2006, the annual NEE was surprisingly similar in other years, varying from –950 to –990 g CO2 m–2 a–1. The estimated uncertainty in the annual budget, including the random measurement and gap-filling error and the uncertainty in the high-frequency loss correction and gap-filling of the longer than 2-day gaps, varied from 35 to 114 g m–2 a–1, corresponding to 3.6 to 22% of the respective annual balance (Appendix B)."

–Fig. 5c, d.

"TECHNICAL CORRECTIONS Page 1: Line 10: turn into (instead of turn to) "

–corrected

"Line 11: We measured the carbon ("the" is missing) "

–added

"Line 11: NEE of CO2 was measured with an eddy covariance method. . . "

–"the" added

"Line 14: Biomass (in singular, not plural) and litter production. . .. "

–ok

"Line 14-16: Consider changing to: Soil balance was estimated. . .from NEE, and this showed that the soil itself was a carbon sink as well. "

–ok, changed

"Page 6: Line 34: Consider changing to: WT varied also spatially ("quite much" should be deleted). "

–ok, deleted

". ... Page 10: Line 33: Consider changing to: "The value is considerably smaller than that reported for agricultural fields." "

–ok, changed

**Anonymous Referee #3**

On my first read of this paper I thought it was an excellent study. I still do but with one major caveat, which I outline below in detail.

The authors are correct that the commonly used EF by the IPCC for drainage of organic soils do not apply to the case of forested peatlands, particularly the drained forests of Finland. The forest management practices there are well done and the drainage is not excessive. Also the ditches are very clean so that also minimizes emissions. The same authors have published aspects of this story before. Their publications and this study support an argument for Tier 2 and 3 methods applied to the Finnish drained forests. The present manuscript develops the argument further that drained mires for forest remain sinks even in a year of extreme drought. This would be an important proof of the continued sink potential. Unfortunately, on my second read I realized that the only year where there are no eddy covariance measurements during the growing season was the year of the significant drought - 2006.

The authors show the large, persistent water table drop – it is far in excess of any other year and the duration of the drought in exceptional unique in their data set. To accept the evidence that is used to support the authors' main conclusion one has to believe that the gap-filled techniques are appropriate for drought conditions. I doubt this very much. The authors evaluate their model for 'normal' years but they provide no evidence that the model applies for the drought conditions experienced in 2006. The actual test is very weak – they examine annual NEP and NECB. It is simulated reasonably well but this is largely irrelevant as the key test is to see if the model simulates GPP and Reco under drought. The reader has to know how well the model performs for extremes, particularly how well its captures the influence of high VPDs and lower water tables. Most gap-filling algorithms do well for average conditions – this is what they are designed to do. However, like most ecosystem models they do very poorly when the conditions deviate beyond the normal range of variance. The water table during 2006 was much deeper than anything experienced during the other three years. Further, the drop in water table is a persistent secular trend. The authors provide no evidence that their gap-filling algorithms can simulate GPP or Reco under these conditions. The model has no dependency on water availability. The GPP model has VPD and PPFD so the reduction in water availability has to come through the VPD function but there is no link. However, it appears that the VPDs experienced in the drought were unique – i.e. well outside the range that was used to develop the parameters in the model. I assume GPmax is derived for normal conditions not drought conditions.

For me to believe their results I would need to see the following:

1, A demonstration that the parameters in their LUE model (GPmax) apply to drought conditions;

2. How Reco is sensitive to changes in water storage, or how temperature reflects the effect of low water contents;

3. the range of conditions that the data was used to fit the gap-filling functions and how for outside that ranges were the conditions in 2006; and

4. an error analysis that looks at the performance of the model functions with extreme conditions – firstly using measurement and simulated data within the range of conditions observed, and then by sensitivity analysis. The sensitivity analysis will not have any moisture effects in it so maybe an examination of the model the authors use versus what other models that incorporate moisture produce.

With these four steps the authors could place the 2006 data on solid ground or determine that their main conclusions are an artifact of the gap-filling approach. Without this the paper is simply infer untested functions for conditions they were never designed to handle and the key argument of the paper collapses.

Given the that central argument of the manuscript is dependent on the 2006 Fluxes and these are not measurements but gap-filled the authors have to provide proof the gapfilling is appropriate. If the authors can provide this proof then this manuscript would be useful addition to the literature and it would have policy relevance. Without the proof of the gap-filling validity the evidence supporting the main conclusion collapses.

**Answers to REF 3:**

The referee is right in pointing out the importance of gap filling and emphasizing the scarcity of measured data in summer 2006. However, we can show that this does not have that dramatic effect neither on the uncertainty of the annual balances nor the conclusion that the site remains as a CO2 sink also over the dry year. As an answer to the criticism related to the long data gaps in 2006 and the resulting implications, we present the following points:

**1. The title will be changed**. We acknowledge that the original title "Carbon accumulation in a drained boreal bog was decreased but not stopped by seasonal drought", which emphasized the dry year (2006) and its C balances, was a bit misleading and we will change it. The actual main messages of the paper are: i) we present the full C balance of the ecosystem and its components, employing several methods and models; ii) the forest ecosystem acted as a quite steady CO2 sink even though it is a drained peatland, and the sink persisted even during a year with an exceptionally dry spring/summer and exceptionally high precipitation in autumn; iii) the drought decreased not only GPP but also respiration. By reformulating the title to '**Persistent carbon sink at a boreal drained bog forest**' we

want to emphasize that our aim was to study the processes that explain this sink capacity during several years, rather than focusing on a single, dry year.

**2. We have plenty of good data from the driest month, i.e. August 2006.** Concerning referee's comment on the missing data during the growing season of 2006 ('the only year where there are no eddy covariance measurements during the growing season was the year of the significant drought – 2006'), we would like to point out that the situation was not that bad. There were long data gaps in June and July in 2006, as we have clearly admitted in many places in the text. However, there were still valid NEE observations in both June and July (225 and 74, respectively). In August and September, when the deepest water table levels took place (Fig. 2), the problems with electricity were overcome and there is a plenty of high-quality EC data available, the monthly data coverage being 50 and 66% in August and September, respectively. Even though the coverage was significantly lower in June and July, the NEE model was still able to produce estimates of Rs0, GPmax and alfa for each month (Fig. 5 in the MS, which was redrawn to include also June and July, see Fig. R1 below). This will replace Fig. 5 in the original MS. To save space the response curves were deleted, and for this reason the related text on p. 7 (lines 19-21) was changed.

[Figure]

Figure 5. Parameter values a) $GP_{MAX}$ and b) $R_{REF}$ ± 95% confidence intervals (see Eqs. A2 and A3 in Appendix A, respectively) for June–October in 2005–2008. For the respiration model, a constant value of $E_0$=200K was used, and the GPP model was used here without the VPD term..

**3. The actual gap-filling model was dynamic and therefore tuned for drought conditions.** The gap-filling algorithm (explained thoroughly in Appendix A) is based on fitting A2 and A3 to the measurement data using a data window of a varying length, to produce daily parameter values. In places with a long gap in the data, parameter values are linearly interpolated. Although the monthly parameters shown in Fig. R1 are not exactly those used in the gap-filling model, they reflect the dynamic and data-derived nature of the model. This means that gap-filling was always adjusted to measured data and the parameters reflected the current conditions as far as possible. As can be seen from Fig. R1, the GPmax parameter was significantly reduced in July, August and September in 2006, most likely because of the drought. Thus the response to water availability does not 'come through the VPD function', as the referee suggests. Similarly, the respiration parameter Rs0, obtained by fitting the temperature-response model to the data, showed reduced values in August (even though the difference to other years was not statistically significant). Thus we can conclude that, because all the fits were done dynamically to measured data, the drought is evidently reflected in the parameter values. However, the long gaps introduce an additional error to the gap-filled fluxes, which is addressed below.

**4. Although the uncertainty in the gap-filled NEE of summer 2006 is high, we can ensure that the annual NEE was negative**. We admit that, due to the long gaps in the NEE data, the uncertainty in the $CO_2$ balance of June-July 2006 is high and have already highlighted this in several places in the original manuscript (p. 7, lines 17-19; p. 10, line 3; p. 33, lines 27-31). Because our gap-filling model fills the long gaps by linear interpolation, the outcome depends on the somewhat 'random' selection of the start and end points of the interpolation. This has already been discussed and taken into account in the error estimate (Appendix B). However, as a further sensitivity test, we estimated the most conservative range for the NEE uncertainty. This was done by assuming different parameter scenarios during the longest gap in the parameters (13 June – 4 August, during which there were only 212 valid NEE observations available). In one scenario, a parameter had the starting point value during the whole gap, while in the other it dropped immediately to the level at the end point and stayed there over the whole gap. The annual NEE was calculated for each combination of different Rs0 and GPmax dynamics. Selecting the two most extreme cases (either both Rs0 and GPmax were reduced for the whole gap, or both were increased) produced an annual NEE of –377 and –844 g $CO_2$ m$^{-2}$ s$^{-1}$, respectively. These can be considered as the most conservative estimates of annual NEE, and thus we can safely conclude that the annual NEE was negative in 2006, i.e. the ecosystem acted as a CO2 sink even during the exceptionally dry year. We will add this sensitivity analysis to the Appendix B "Uncertainty analysis of NEE".

**5. The autumn respiration was exceptionally high in 2006.** It is important to note that the higher annual NEE (lower uptake) in 2006 was not solely due to the reduced GPP and Reco during the summer but also was strongly influenced by the clearly higher respiration in autumn. The period of low precipitation continued in September, and the soil temperatures (particularly at 30 cm depth) were high in September and October. The high rainfall in October, combined with high soil temperatures and reduced respiration in the previous months resulted in very high respiration rates for the rest of the year (Fig. 4b). We have already pointed this out in the original manuscript on p.7, lines 16 and 26–28:

'The higher RECO in autumn months after the drought and heavy rains in October (Fig. 2) furthermore increased the difference to other years: the cumulative NEE in October-December in 2006 was 320 g $CO_2$ m–2, while in other years it varied from 130 to 190 g $CO_2$ m–2.' The main message here is that the lower NEE in 2006 cannot be attributed only to the summer when the amount of measured flux data was low, as clearly explained in the original manuscript.

[revised manuscript text omitted]